# Field-induced compensation of magnetic exchange as the possible origin of reentrant superconductivity in UTe$_2$

Toni Helm [1,2] ✉, Motoi Kimata [3], Kenta Sudo[3], Atsuhiko Miyata[1], Julia Stirnat[1,4], Tobias Förster[1], Jacob Hornung[1,4], Markus König[2], Ilya Sheikin [5], Alexandre Pourret [6], Gerard Lapertot [6], Dai Aoki[7], Georg Knebel [6], Joachim Wosnitza[1,4] & Jean-Pascal Brison [6]

The potential spin-triplet heavy-fermion superconductor UTe$_2$ exhibits signatures of multiple distinct superconducting phases. For field aligned along the $b$ axis, a metamagnetic transition occurs at $\mu_0 H_m \approx 35$ T. It is associated with magnetic fluctuations that may be beneficial for the field-reinforced superconductivity surviving up to $H_m$. Once the field is tilted away from the $b$ towards the $c$ axis, a reentrant superconducting phase emerges just above $H_m$. In order to better understand this remarkably field-resistant superconducting phase, we conducted magnetic-torque and magnetotransport measurements in pulsed magnetic fields. We determine the record-breaking upper critical field of $\mu_0 H_{c2} \approx 73$ T and its evolution with angle. Furthermore, the normal-state Hall effect experiences a drastic suppression indicative of a reduced band polarization above $H_m$ in the angular range around 30° caused by a partial compensation between the applied field and an exchange field. This promotes the Jaccarino-Peter effect as a likely mechanism for the reentrant superconductivity above $H_m$.

Superconductivity is notoriously fragile under magnetic field, all the more when the superconducting critical temperature is small. However, the sensitivity of superconductors to magnetic field is influenced by a variety of factors. For example, a whole class of strongly correlated electron systems called "heavy fermions" exhibits critical fields several orders of magnitude larger than other superconducting systems with similar $T_c$ (usually sub-Kelvin), precisely because the quasi-particles possess heavy effective masses, or equivalently, very slow Fermi velocities[1–3]. In many heavy-fermion materials, the upper critical field is limited at low temperatures by the paramagnetic limit that arises from the Zeeman coupling of the Cooper pair spins to the external field[4,5]. In other

superconductors, only a strong 2D character may allow for enhanced upper critical fields close to that limit.

The recent discovery of superconductivity (SC) in the heavy-fermion metal UTe$_2$[6] with a critical temperature $T_c \approx 2$ K, triggered much excitement, as its critical field reaches values approaching those of high-Tc superconductors. Moreover, it appeared very quickly as a potential candidate for topological spin-triplet SC[6,7] with multiple unconventional superconducting phases under field or pressure[8–19]. Spin-triplet SC is a rare phenomenon, expected to arise as a consequence of magnetic fluctuations in strongly correlated materials. It is characterized by a particularly high stability against external magnetic fields due to the suppression of Pauli depairing. Indeed, a key

[1]Hochfeld-Magnetlabor Dresden (HLD-EMFL) and Würzburg-Dresden Cluster of Excellence ct.qmat, Helmholtz-Zentrum Dresden-Rossendorf, 01328 Dresden, Germany. [2]Max Planck Institute for Chemical Physics of Solids, 01187 Dresden, Germany. [3]Institute for Materials Research, Tohoku University, Sendai, Miyagi 980-8577, Japan. [4]Institut für Festkörper- und Materialphysik, Technische Universität Dresden, 01062 Dresden, Germany. [5]Laboratoire National des Champs Magnétiques Intenses (LNCMI-EMFL), CNRS, UGA, 38042 Grenoble, France. [6]Univ. Grenoble Alpes, CEA, Grenoble-INP, IRIG, PHELIQS, 38000 Grenoble, France. [7]Institute for Materials Research, Tohoku University, Oarai, Ibaraki 311-1313, Japan. ✉e-mail: t.helm@hzdr.de

characteristic of $UTe_2$ is an anisotropic upper critical field, $H_{c2}$, that exceeds the paramagnetic limit along all field orientations[6,20]. In particular, SC survives up to a metamagnetic transition at approximately $\mu_0 H_m = 35$ T, for field oriented along the magnetically hard $b$ direction[12,19,20]. These findings resemble those reported for ferromagnetic superconductors, such as UCoGe and URhGe[21,22]. More surprising, the compound is able to reestablish SC even at higher fields, just above $\mu_0 H_m$ at ~40 T for field oriented at $\theta \approx 30°$ away from $b$ towards the $c$ axis[12].

The new reentrant high-field superconducting phase (from here on referred to as hfSC phase) appears to extend into an extreme field range beyond 60 T, with a yet-to-be-determined $H_{c2}$ [12,17]. The nature of the superconducting ground state, the identification of the different field or pressure-induced superconducting phases, and their relation to topological SC are still under debate, notably from a theoretical point of view[23–26]. The mechanisms behind this record-breaking hfSC phase and its relation to the low-field superconductivity (lfSC) are a puzzle and one of the key questions to solve.

Indeed, little is known about the mechanisms responsible for the high-field superconducting phases: neither is it clear how exactly SC is suppressed for $H\|b$ once $H_m$ is approached; nor why SC can reestablish for field orientations near ≈ 30° within the $(b, c)$ plane above $H_m$. Hall-resistivity measurements with $H\|b$ revealed a significant anomalous Hall effect (AHE), which has been associated with coherent skew scattering that dominates the electrical-transport below $T \approx 20$ K[27]. A sign change in the ordinary Hall coefficient and thermoelectric power, and a discontinuity in the $T^2$ term of the temperature-dependent resistivity or the specific heat at $H_m$ indicate a strong impact of the metamagnetism on the electronic band structure and on the correlations[17,19,27,28].

The field-reinforced SC observed for $H\|b$ below $H_m$ [6,20] is associated with an enhancement of magnetic fluctuations in the vicinity of the metamagnetic transition[12,19,20,29–31]. Although $H_m$ represents the limiting scale for SC for $H\|b$, it is also the enabling lower barrier for the hfSC phase. This suggests that magnetic interactions connected with $H_m$ play a key role for the emergence of the field-enhanced and reentrant SC of $UTe_2$. At low temperature, the metamagnetic signature is a step-like change in the magnetization[12,28,31] and in various transport properties such as the residual resistivity[14,17,29], and the Hall effect[27]. The metamagnetic transition is sensitive to the field alignment[12,17,28]. It shifts to higher fields upon changing the field orientation either from $b$ to $c$ or from $b$ to $a$. However, the jump of the magnetization at $H_m$ seems to remain unaffected by an orientation change of 30° within the $(b, c)$ plane[12,28,31]. Presently, the only quantity that differs is the sign of the specific heat jump at $H_m$, negative for $H\|b$[19,28], but becoming positive at 30°[28]. Interestingly, pressure-dependent investigations have revealed that the hfSC phase is not necessarily tied to $H_m$: at large enough pressures, hfSC emerges at field values larger than $H_m$[32].

Here, we present studies of magnetic torque, magnetoresistance, and Hall effect in pulsed magnetic fields up to 70 T for micron-sized samples. They are cut from single crystals of $UTe_2$ by focused-ion-beam (FIB) microfabrication. This enables us to perform measurements in pulsed magnetic fields with enhanced precision in a rather noisy environment compared to steady magnetic fields. We trace the metamagnetic and superconducting transitions in the $(b, c)$ plane. We confirm the emergence of hfSC phase around $\theta = 30°$ at fields above 40 T. We extrapolate the maximum upper critical field to $\mu_0 H_{c2} \approx 73$ T and determine its variation with angle. We trace the magnetic torque through $H_m$ and demonstrate that the spins reside in a non-collinear configuration with a dominant $b$-axis component. Furthermore, we show that the high-field Hall coefficient, having an orbital and a significant AHE component, experiences a drastic suppression as the field orientation approaches the hfSC region around 30°, even though magnetization, magnetic torque and magnetoresistance remain finite.

We propose a new interpretation of the AHE at low temperature in the polarized phase of $UTe_2$ above $H_m$, which suggests a scenario connecting the suppression of the AHE around 30° and the emergence of the hfSC phase. It relies on a field-induced enhancement of the pairing strength together with an angle-dependent band polarization.

## Results and discussion

We investigated several micron-sized samples produced from one oriented single crystal with a superconducting $T_c$ of 1.6 K. The micromachining was performed by means of Ga or Xe FIB systems (for details, see the methods section). This FIB approach enables precise geometries suitable for microcantilever-torque experiments on magnetic materials with strong torque responses as well as high-precision electrical-transport measurements on metallic (i.e., highly conductive) materials with current running along any desired direction (see images in Fig. 1a, b). In this work, we will present results obtained for three transport devices shaped in the standard Hall-bar geometry. Additional torque and magnetotransport data are provided in Supplementary Notes S1–S3. A preliminary characterization of the zero field resistivity in micron-sized structures yielded no significant differences compared to results reported for bulk samples, see Supplementary Fig. S2. The critical temperature of 1.6 K is not altered by the fabrication in comparison to the bulk sample, and the overall temperature dependence is reproduced.

### Magnetic torque around the metamagnetic transition

We investigated the isothermal magnetic torque of $UTe_2$ by means of microcantilever torque magnetometry (see Fig. 1a) in pulsed fields up to 70 T for various angles. This technique probes the magnetic anisotropy and complements magnetization measurements[33]. As a consequence of the step-like increase of magnetization and the change in anisotropy of $UTe_2$ at $H_m$[31,34], the response in magnetic torque is strong. Thanks to the sample preparation by FIB, the volume of the sample is small enough to limit the maximum torque to a safe value preventing damage to the microcantilever.

Figure 1a presents torque data recorded at 0.7 K. An additional data set for $T = 1.5$ K can be found in the Supplementary Information Fig. S1. The tilt angle $\theta$ was varied between $H\|b$, i.e., $\theta = 0°$, and the $c$ direction. The metamagnetic high-field transition shows up as a step-like feature at fields above 35 T. The monotonic change in $\tau(H)$ at constant angle reflects that of the bulk magnetization. It confirms that besides the jump at $H_m$, there are no other anomalies in the magnetization for all the measured angles within the $(b, c)$ plane. Interestingly, the jump in $\tau(\theta)$ depending on the tilt angle exhibits a pronounced local minimum at $\theta \approx 25°$, for all fields above $H_m$ in this angular range. This is best seen when we plot the torque magnitude against the tilt angle, at low temperature, see Fig. 1d and Supplementary Fig. S1. As the magnetic torque reflects the magnetic moment of the anisotropic crystal, it is sensitive to the magnetization component perpendicular to the magnetic field. Its maximum is expected around 45°, consistent with our data. The noticeable drop at 25°, therefore, is indicative of a bulk feature in the magnetic part of $UTe_2$ around this angular range coinciding with the reentrant high-field superconducting phase. However, at higher temperature than $T = 1.5$ K the feature seems absent, as can be seen in Supplementary Fig. S1. Note: Both the low-field and the high-field SC is not discernible in the pulsed-field torque data. This may be a consequence of the fast $dH/dt$. The observed, drop of the torque around 25°, however, may originate from a screening associated with superconducting diamagnetic currents. More work is required to pinpoint the origin of the decrease of the torque in this angular range.

Remarkably, the jump of the torque at $H_m$ for finite tilt angles changes from strictly negative to strictly positive values, not from negative values to zero. Therefore, in the "polarized state" above $H_m$

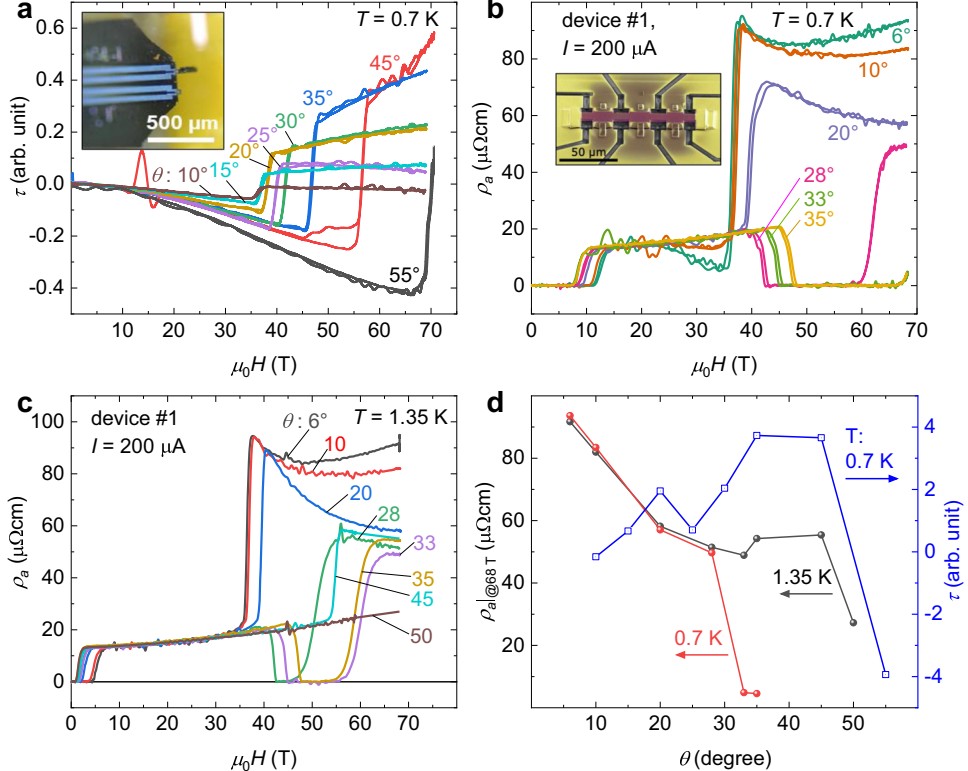

**Fig. 1 | Magnetic torque and magnetoresistivity of UTe₂. a** Magnetic torque vs. pulsed magnetic field for various angles recorded for a thin sample (90 × 15) μm at a temperature of 0.7 K. The tilt angle, θ, denotes the field orientation in the (b, c) plane, where 0° corresponds to H∥b. Inset: Picture of the piezoresistive micro-cantilever with a lamella-shaped sample attached to it. **b, c** Resistivity vs. pulsed magnetic field for device #1 recorded at T = 0.7 and 1.35 K, respectively, for various

tilt angles. Inset of (**b**): False-color scanning-electron-microscope image of the FIB structured Hall-bar device #1 with a thickness of 2 μm and I∥a. The b axis points along the normal of the substrate. **d** First layer: Resistivity at 68 T from the data in (**b**) and (**c**) versus angle at 0.7 and 1.35 K. Second layer: Magnetic torque at 60 T from data in (**a**) versus angle at 0.7 K.

the magnetic moments and H are not collinear and have a dominant b-axis component even for θ ≥ 45°. This is an additional feature revealed by our magnetic torque measurements. In comparison to the previous magnetization studies[12,31], magnetic torque is sensitive to the transverse component of the magnetization.

## High-field superconductivity and its electrical transport signature

We conducted resistivity and Hall-effect measurements in fields up to 70 T. Isothermal resistivity curves recorded for Hall-bar device #1 (see inset in Fig. 1b) at 0.7 and 1.3 K, with field oriented along the b axis, are presented in Fig. 1b and c, respectively. The in-plane resistivity $\rho_a$ exhibits a step-like change at the metamagnetic transition that sets in at $\mu_0 H \approx 35$ T for H∥b. This feature is consistent with the metamagnetic jump at $H_m$ in magnetic torque. We provide additional data recorded for device #3 at various temperatures ranging between 1.4 and 77 K for the fixed field orientation H∥b in Supplementary Fig. S2. Upon decreasing temperature, the metamagnetic transition evolves from a broad anomaly into a sharp first-order type transition. Such an evolution resembles the behavior observed in other heavy-fermion metals with metamagnetism in high fields[35–37]. Regarding the hfSC phase, our results are in line with previous reports[12,17]. However, we show its extent to higher fields with a far improved resolution.

First, we focus on the data recorded for orientations close to H∥b: In the 6° curve we observe a fingerprint of the reentrant behavior of the lfSC phase reported previously[6,20]: the normal state is reached above 12 T until the resistivity starts dropping again above 20 T, see Fig. 1b. Apparently, the reentrant signature is suppressed in the 1.35 K data,

shown in Fig. 1c. As we increase θ to 20° and above, the magnetoresistance in the normal state below $H_m$ remains unchanged. Above $H_m$, it gradually evolves from a positive upturn into a monotonic change. Similar to our observations in magnetic torque, the metamagnetic transition shifts towards higher fields. However, in the case of resistivity, the strong step-like feature is quenched by the onset of zero resistance associated with an additional reentrant superconducting phase that sets in, once θ reaches beyond 20°. At 0.7 K, the resistivity curve for the highest tilt angles θ = 35° tested, still exhibits SC that extends up to 69 T. In comparison, for the same angle but at 1.3 K, the resistance reaches the normal state again already at fields below 60 T. At 1.35 K, no trace of hfSC was discernible for angles from 45° onward, see Fig. 1c. At 45°, we observe a step-like resistance increase followed by a negative slope as for angles below 28°. For θ = 50°, $H_m$ is pushed above the field range accessed in this experiment. Hence, the normal-state resistance increases monotonically up to the highest field. As can be noticed from Fig. 1d, the magnetoresistivity $\rho_a(\theta)$ in the normal state above $H_m$ experiences a slope change from positive to negative upon rotation away from H∥b. The overall amplitude at 68 T exhibits a dip near 30°, the angle where the hfSC appears to be the strongest.

In Fig. 2a, we present a data set of the resistance recorded for various temperatures between 0.7 and 1.4 K at the fixed orientation θ = 35°. The critical fields of the lfSC and the hfSC phases at different temperatures were determined from the inflection points of the magnetoresistance curves. At the lowest temperature reached in our experiment, T = 0.7 K, the superconducting phase survives magnetic fields close to 70 T. Its onset appears to be directly pinned to the metamagnetic transition.

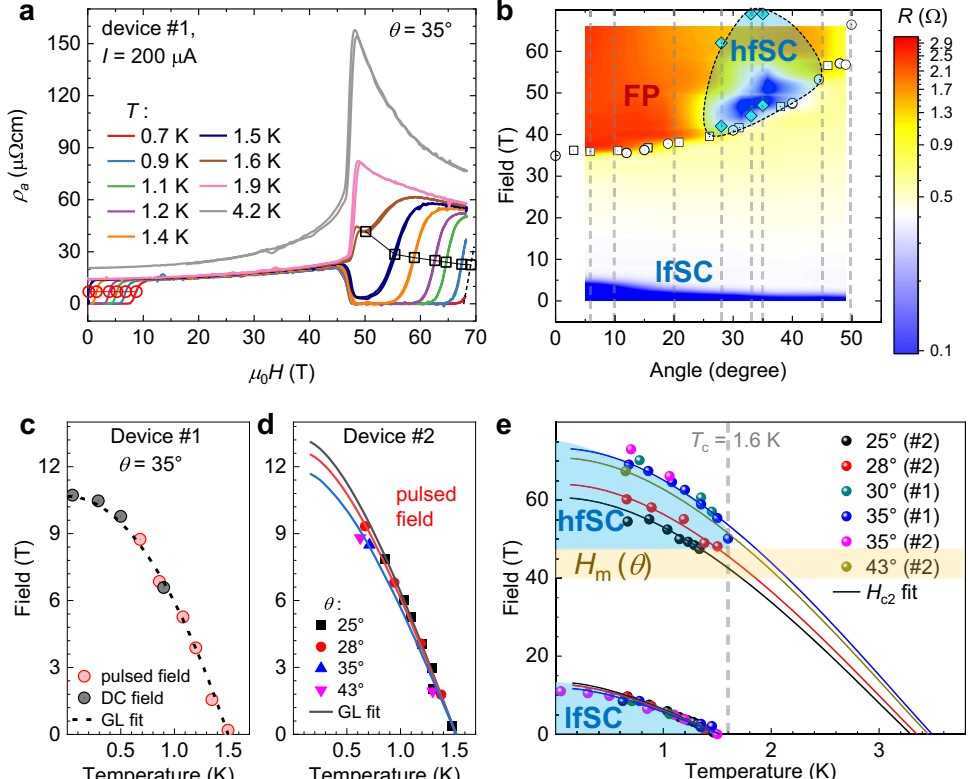

**Fig. 2 | High-field phase diagram of UTe₂. a** $a$-axis resistivity versus magnetic field at fixed tilt angle of $\theta = 35°$ for various temperatures. **b** Contour plot created from data presented in Fig. 1b, c. White squares and circles mark the metamagnetic transition field measured by pulsed-field torque magnetometry presented in Fig. 1a and Fig. S1. Cyan diamonds indicate the superconducting transition fields in the 0.7 K data set presented in Fig. 1b. The black dashed line indicates the approximate extension of the SC region at 0.7 K. **c, d** Temperature dependence of the

superconducting critical field of the lfSC phase, determined for device #1 and #2, respectively, in pulsed and steady fields. Dashed and solid lines are GL fits (for details see Supplementary Note 5). **e** Critical fields of the hfSC and lfSC phase determined in pulsed field. Solid lines are fits of $H_{c2}$ in the pure orbital limit, using a strong-coupling constant $\lambda = 1.51$, 1.53, 1.58, and 1.567, respectively, at 25°, 28°, 35°, and 43°.

## $H_{c2}$ in the field-induced reentrant hfSC phase

Figure 2b shows a schematic phase diagram comprised of a contour presentation of the data from Fig. 1c and the transition fields $H_m$ and $H_c$ determined from our torque and resistance results.

Figure 2c shows the superconducting $H_{c2}$ of the lfSC phase for device #1 determined in DC (gray) and pulsed (red) magnetic fields oriented parallel to the $b$ axis (squares) and tilted 30° (circles) towards the $c$ axis within the $(b, c)$ plane. Figure 2d shows similar data for device #2 for fields applied at different angles within the $(b, c)$ plane, measured all in pulsed fields.

For spin-singlet superconductors, $H_{c2}$ has an upper limit fixed by Pauli paramagnetism[4,5]. The limiting field, $H_{Pauli}$, for a singlet superconductor at 0 K can be approximated by $\mu_0 H_{Pauli} \approx \sqrt{2}\Delta/(g\mu_B) = 1.86[\text{T/K}] \cdot T_c$, valid in the BCS weak-coupling limit without any spin-orbit interaction and for a free-electron value of the $g$-factor: $g = 2$. In the case of UTe₂, this would roughly lead to 3 T, much smaller than the measured critical fields (reaching close to or beyond 10 T in all directions). A combination of spin-triplet SC, strong superconducting coupling, and strong spin-orbit interactions could be responsible for this violation of the paramagnetic limit in all field directions[38].

The evolution of $H_{c2}$ with temperature in the lfSC phase for the field tilted by approximately 35° towards the $c$ axis follows the standard (close to parabolic) temperature dependence of $H_{c2}$ in the pure orbital limit. Fits of the data were done in the strong-coupling regime appropriate for UTe₂ [39], using a moderate value of the strong-coupling constant of $\lambda = 1$ (solid lines in Fig. 2c, d). In the Ginzburg-Landau weak-coupling regime, the slope of $H_{c2}$ at

$T$ is given by ref. 40:

$$\frac{dH_{c2}}{dT} \approx 9\Phi_0 \left(\frac{k_B T_c}{\hbar \langle v_F \rangle}\right)^2 \frac{1}{T_c}. \tag{1}$$

Once $\lambda$ and $T_c$ are fixed, the slope of $H_{c2}$ at $T_c$ (hence, the average Fermi velocity perpendicular to the applied field $\langle v_F \rangle$) is the only parameter left to determine the complete temperature dependence of $H_{c2}$ in this approximation. From the best fits, we find 6700 m/s ≤ $\langle v_F \rangle$ ≤ 7100 m/s for angles between 25° and 35° in the $(b, c)$ plane.

Let us now turn to the critical fields of the hfSC phase, above $H_m$. The points shown in Fig. 2e were determined in the hfSC phase for two devices, again at various tilt angles. In our pulsed-field setup, we were limited to temperatures above 0.7 K. A prerequisite to a profound analysis of $H_{c2}$ in this phase is a theoretical model explaining the mechanisms for reentrant SC above $H_m$. Indeed, in the likely case of a connection between hfSC and magnetic fluctuations that develop upon approaching $H_m$, we can expect a reduction of the pairing strength (following the observed reduction in the specific heat), $\lambda$, once the external magnetic field becomes much larger than $H_m$. Such a behavior is reminiscent of that observed for the field-reinforced SC for $H \| b$ below $H_m$ [19], where the coupling strength increases on approaching $H_m$. However, to date, a well-defined theoretical scenario for the field dependence of $\lambda$ in the hfSC phase is lacking.

We will discuss a proposal for such a model later in the paper. In order to determine minimal constraints from the data, we first analyze them without any field dependence of $\lambda$. We use the same strong-coupling model proposed for fields below $H_m$ in ref. 19 in combination

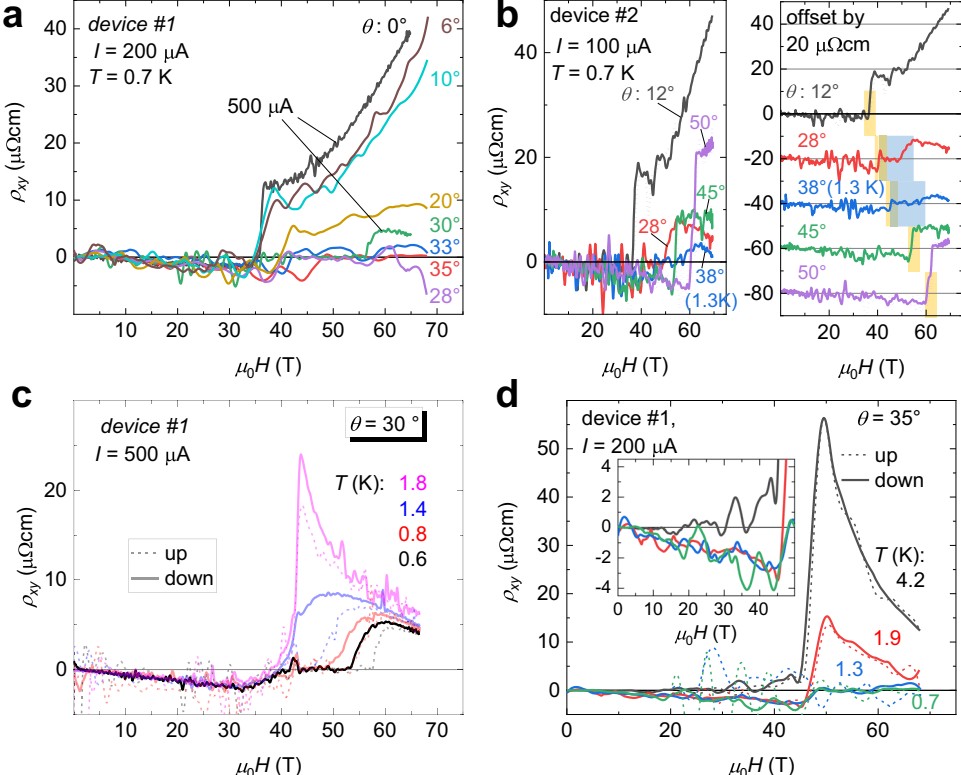

**Fig. 3 | High-field Hall effect of UTe₂.** Hall resistivity of device #1 recorded with two different currents for field-up and -down sweeps (**c**), (**d**) at fixed field orientations, θ = 30° and 35°, for various temperatures, and (**a**) at fixed temperature, T = 0.7 K, for various angles. **b** Left panel: Hall resistivity recorded for device #2 with I = 100 μA at fixed temperature and different angles. Right panel: The curves were shifted by a constant offset for better visibility. H_m and the hfSC region are highlighted by yellow and blue shaded bars, respectively. Inset in (**d**): zoom into the region below 50 T.

with the hypothesis that the paramagnetic limit is absent for the hfSC phase (as for a spin-triplet equal-spin-pairing (ESP) state). λ is adjusted in order to have a large enough $T_c$ (in zero field) that could explain the survival of SC above $H_m$. The orbital limit is mainly controlled by an (Fermi surface) averaged renormalized Fermi velocity $\langle v_F \rangle$, directed perpendicular to the magnetic field. The renormalization includes the effect of the pairing interactions. Hence, $\langle v_F \rangle$ can be written as $\langle v_F \rangle = \frac{\langle v_F^{band} \rangle}{1+\lambda}$, where $\langle v_F^{band} \rangle$ is a bare "band" averaged Fermi velocity (renormalized by all interactions but the pairing interaction), for which we used the same values along the $b$ and $c$ axis as in the low-field phase[19] (see Supplementary Note 5 for more details on the model). The required values of λ range from 1.51 (at 25°) to 1.58 (at 35°), and the corresponding fits are shown in Fig. 2e.

Remarkably, we could use the same $\langle v_F^{band} \rangle$ as a control parameter of the orbital limit for the lfSC and hfSC phases at the same angle. It seems to imply that correlations (except for the change of the value of λ) and the Fermi surface experience no dramatic change at $H_m$. In other systems, where quantum-oscillation measurements could be performed, such as the well-documented case CeRu₂Si₂[41,42] as well as the uranium systems UPt₃[43] and UPd₂Al₃[44], Fermi surface changes were observed across the metamagnetic field $H_m$, as well as heavy masses just above $H_m$. However, these heavy masses should be suppressed much faster by external magnetic field in cerium-based systems, which show a clearer trend to localization of the *f*-electrons under field and smaller Kondo temperatures than uranium systems. We will discuss later particular aspects of UTe2 that explain why it preserves large effective masses above $H_m$, at least for the singular field orientations where SC reappears. From this first analysis, we conclude that the existence of the hfSC phase still requires an absent paramagnetic limit and large effective masses similar to the lfSC phase (same $\langle v_F \rangle$). This, together with the enhanced (zero-field) critical temperature, is

sufficient to explain that SC can survive at these record high fields. Our approach reproduces the overall temperature dependence of $H_{c2}$ reasonably well and yields $\mu_0 H_{c2}^{max} \approx 73$ T (±1 T) between 30 and 35°. The obtained $T_c$ values, extrapolated to zero field, range between 3.2 and 3.6 K. In Fig. S4 we present a normalized comparison of $H_{c2}(\theta)$ for both the hfSC and the lfSC phases. The remarkable anisotropy of $H_{c2}$ in the hfSC with a peak around θ ≈ 35° is contrasted by the monotonically decreasing $H_{c2}$ of the lfSC. The maximum $H_{c2}$ value sets a record-breaking mark for SC emerging in a heavy-fermion compound to date. The existence of heavy quasiparticles at fields above 40 T means that renormalization of the effective masses by the Kondo effect is still effective above $H_m$.

## Strong suppression of the Hall effect in the vicinity of the hfSC phase

In Fig. 3a–d, we present Hall-resistivity data recorded in pulsed magnetic fields for devices #1 and #2 at two different currents and for various angles and temperatures. The Hall resistivity is composed of the ordinary component linked to the charge-carrier density and mobility, and an AHE component, whose origin is still the subject of intense research[45]: it may have an intrinsic origin related to the topology of the electronic band structure, well identified in ferromagnets, or an extrinsic origin arising from different scattering mechanisms (skew scattering or side-jump), all a consequence of spin-orbit interactions.

In the case of heavy-fermion systems, even though there is no accepted complete microscopic theory[45,46], the most successful interpretation of the AHE relies on skew scattering from local and itinerant *f*-electrons[46,47]. An analysis of the electrical-transport coefficient obtained in steady fields up to 35 T by Niu et al. pointed out that coherent skew scattering of the conduction electrons is the dominant

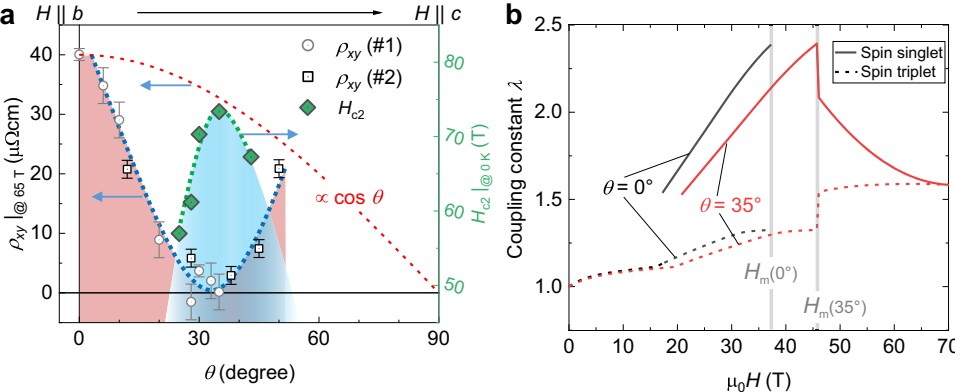

**Fig. 4 | Evidence for the Jaccarino-Peter compensation effect in UTe₂. a** Angular dependence of the normal-state Hall resistivity at 65 T and 0.7 K of devices #1 and #2, taken from Fig. 3a, b. The blue dashed line is a guide to the eye that highlights the observed strong suppression around $\theta \approx 30–35°$. Green diamonds are $H_{c2}$ values extrapolated to zero temperature, taken from Fig. 2e. The red dashed line follows $\cos\theta$. **b** Strng-coupling constant $\lambda$ used for describing the experimental $H_{c2}(T)$ data points presented in Fig. 2e, within the two proposed scenarios: Solid (dashed) lines are with (without) paramagnetic limitation of $H_{c2}$, assuming a spin-singlet (spin-triplet-ESP) state. Below $H_m$ at 0° ($H\|b$), the curves are those from ref. 19. Below $H_m$ at 35°, the curves are scaled by $1/\cos(\theta)$, assuming that only the $b$ component $H$ determines $\lambda(H)$. The field dependence of $\lambda$ above $H_m$ is based on the compensation mechanism in both cases as explained in the main text and in Supplementary Note 5. The spin-singlet result resembles the $H$ dependence reported for the specific heat (for $H\|b$) or the $A$ coefficient around $H_m$ [19,29–31].

contribution to Hall effect below about 20 K for $H\|b$ [48]. We provide additional Hall data recorded at temperatures between 1.4 and 77 K for device #3 in the Supplementary Information (Fig. S2), consistent with the previous report[48].

In the following, we will focus on the angular dependence of the Hall effect in UTe₂. We recorded high-resolution Hall-effect data for two different transport devices with $I = 500\,\mu A$, $200\,\mu A$, and $100\,\mu A$. The lower currents provide the least heating of the samples but a reduced signal-to-noise ratio. The overall low-field Hall signal acquires a negative magnitude and slope once the normal state is reached. At $H_m$ a sharp jump, similar to that of the magnetoresistivity, occurs and the Hall resistivity changes sign consistent with observations for $H\|b$ [48]. We observe a drastic change of the high-field Hall signal with angle, see Fig. 3a, b. The overall slope of $\rho_{xy}(H)$ changes from positive to negative at highest fields as we increase the tilt angle to about 28°. This is supported by the high-resolution (larger bias current) measurements of $\rho_{xy}(H)$ shown in Fig. 3c and the higher-temperature measurements shown in Fig. 3d. Moreover, the magnitude of $\rho_{xy}$ becomes strongly suppressed with a hardly discernible jump at $H_m$ (68 T at 38°). Interestingly, at 45° and beyond, the feature at $H_m$ is visible again.

Most importantly, at angles ranging from 28° to 38° the Hall resistivity is zero in the superconducting state just as the resistivity. This is most apparent in the high-resolution data recorded at $\theta = 30°$, with a current of 500 $\mu A$, shown in Fig. 3c. Even though there is a slight difference in the transition field between the up and down sweep, potentially originating from heating, all curves show a zero signal in the superconducting state. Previous pulsed-field resistivity and magnetization studies have already reported a zero-resistance state indicating the hfSC phase[12]. Nevertheless, a low resistivity may indicate a very metallic state, but may not be unambiguous proof for the presence of SC. Here, measurements of Hall resistivity can be of great help: They are sensitive to the nature and to the density of states near the Fermi level mainly responsible for the transport properties. This has been well demonstrated in layered delafossite compounds, where a super-low-resistive ground state was observed with a resistivity at 4.2 K below 0.01 $\mu\Omega$cm (very hard to detect for bulk devices)[49–51]. In this particular case, a large mean free path reduces scattering, resulting in a hardly detectable resistivity response. Yet, the Hall resistivity remains non zero, signaling a well-established Fermi surface. Therefore, the vanishing of the Hall resistivity (within the noise) observed in the hfSC phase for UTe₂, provides further proof for condensation of charge carriers in a superconducting state in the hfSC phase. Note 1: At higher

temperatures, $\rho_{xy}$ below $H_m$ gradually changes from negative to positive and crosses zero (see 4.2 K data in the inset of Fig. 3d). Our resolution of a few micron thin device enables us to distinguish the weak negative low-field Hall effect from the zero signal in the hfSC region, best demonstrated in Fig. 3c. Note 2: in the low field regime, the resolution of $\rho_{xy}$ is degraded due to the small signal amplitude in the normal state. This, together with the increased heating at the end of the magnetic field pulses prevents the observation the effect of SC on $\rho_{xy}$ in the lfSC phase as clearly as in the hfSC phase.

The steep angular suppression of the high-field Hall effect signal is shown in Fig. 4. Therein, we plot $\rho_{xy}$ at a fixed magnetic field of 68 T against the tilt angle. For angles above approximately 40°, $\rho_{xy}$ in the normal state recovers again and reaches values close to those expected from a conventional $\cos\theta$ scaling behavior, indicated by the red-dashed line. The mechanism behind the drop in $\rho_{xy}(\theta)$ is a puzzle, particularly when compared to previous work that explored the electronic properties of UTe₂ at field orientations around 30° tilt within the $(b, c)$ plane. Indeed, previous magnetization measurements observed no significant change of the magnetization jump at $H_m$ for fields along the $b$ axis and around 30° [12,34]. Similarly, the resistivity does not show significant changes around $H_m$ for both field orientations[17], indicating that neither the elastic nor the inelastic scattering display a considerable evolution with angle. Therefore, we expect the AHE component, which is directly proportional to $M$ and to the resistivity or the square of the resistivity, to remain (roughly) constant with angle.

Recent dHvA studies, which confirmed the Fermi-surface topology predicted by band structure calculations[52,53], may hint at specific properties linked to the $\theta \approx 30°$ field orientation. In particular, the warping of the cylindrical Fermi surfaces could meet the so-called "Yamaji magic-angle" condition[54] that can induce a suppressed conduction for a particular field orientation and, thus, affects the density of states at the Fermi edge. To date, the exact Fermi-surface topology in the high-field regime above $H_m$ has not been revealed. Thus, the potential influence of Fermi-surface anomalies on the Hall coefficient above $H_m$ is unknown.

**Analysis of the Hall effect and connection with the hfSC phase**

In the following, we propose a scenario for the reentrant hfSC phase, supported by the analyses of our high-field torque and Hall-effect results. We will show that the origin of the Hall effect above $H_m$ should be revisited, arising most likely from an intrinsic topological contribution. Hence, suppression of the Hall effect is best explained by

that of the band polarization, leading naturally to a Jaccarino-Peter (JP) mechanism for the hfSC phase.

In the Supplementary Information (Fig. S5), we present an analysis of the Hall data along the lines of Niu et al.[48]. Under the assumption that skew scattering (directly proportional to the product $\rho_{xx}^2 M$) is the dominating extrinsic component at low temperatures and with the inclusion of already published magnetization data[12,31], we can extract the normal (orbital) Hall coefficient $R_H$ at $\theta = 30°$ from the intercepts in Fig. S5 (second and third column). Apparently, $R_H$ jumps by a factor of two, i.e., $0.05\,\Omega\text{cm}/T \to 0.1\,\Omega\text{cm}/T$ when transitioning from below to above $H_m$. Intriguingly, the high-field value is almost one order of magnitude smaller than what was reported for the $H\|b$ orientation[27]. This analysis implies, however, that the proposed[27] strong suppression (by a factor 10) of the charge-carrier density for $H\|b$ is not present anymore for tilted field. Moreover, significant changes in the charge-carrier density at $H_m$ have not been confirmed by any other reported quantity, such as the specific heat[19,30,31] or the $A$ coefficient of the resistivity[17]. A dramatic suppression of the density of states would also be hard to reconcile (if persistent for tilted fields) with the appearance of the hfSC phase. We, therefore, argue that the conventional interpretation in terms of normal and skew scattering dominated contributions to the Hall effect proposed in ref. 48 does not hold in the case of UTe$_2$. Indeed, the general understanding of mechanisms behind the AHE has significantly improved in recent years[45,55]. In particular, the role of intrinsic (topological) contributions, expected to scale with $\rho_{xx}^2 M$, has been discussed[45,55]. Such contributions depend on topological invariants associated with the band polarization. They are already present in zero field for ferromagnetic systems. The band structure of UTe$_2$ may host topological features such as Weyl nodes near the Fermi edge[25]. Such contributions should dominate the AHE when the resistivity is in the range between 1 and $100\,\mu\Omega cm$[45]. Moreover, the dependence on magnetization in ferromagnets arises not from magnetic interactions, but simply from the domain alignments: In other words, if such a contribution appeared above $H_m$ due to a sudden band polarization at the metamagnetic transition, it would keep a $\rho_{xx}^2$ dependence, but the $M$ factor might be meaningless. Hence, the strong negative drop of the normal Hall coefficient reported by Ref. 48 can also be explained by the emergence of a strong intrinsic anomalous Hall effect (iAHE) at $H_m$. Furthermore, the role of skew scattering could have been largely overestimated. As a consequence, our observed angle-dependent suppression of the Hall effect around $\theta \approx 30°$ in the $(b, c)$ plane should then reflect the suppression of this iAHE. With an almost angle-independent jump in the magnetization at $H_m$ (at least within the angular range, where hfSC exists[12,31]), the steep decline of the iAHE contribution suggests a suppressed influence of the topological aspect in the band structure on the AHE.

Band splitting with avoided level crossing is key for this intrinsic contribution to the Hall effect[45]. So an appealing possibility is that the suppression of the iAHE contribution arises from a strong decrease of the band polarization in this angular range. It could result from a compensation between the applied field and an exchange field between the conduction bands and local magnetic moments, polarized by the metamagnetic transition. The background picture for this scenario is that a main contribution to the magnetization of UTe$_2$ arises from localized $5f$-electrons. This is consistent with the large nearest-neighbor distance, far exceeding the Hill limit[56]. It is furthermore supported by band structure calculations that predict a Fermi surface dominated by Te-$5p$ and U-$6d$ electrons (partly hybridized with U-$5f$), with at most only small $5f$-electron pockets[26]. These, however, have not been observed by experiments to date[52]. In such a scheme, the jump of the magnetization at $H_m$ arises mainly from local moments having (antiferromagnetic) exchange coupling with the conduction bands, a very natural scheme for a Kondo system. A reduction of the band polarization, arising from the compensation between exchange and applied field above $H_m$ also explains why we can fit $H_{c2}$ in the hfSC

phase with the assumption of unaltered $\langle v_F^{band}\rangle$ values as compared to the lfSC phase (Fig. 2e.): the main effects of the metamagnetic transition on the Fermi surface then disappear. More importantly, this compensation between $H$ and the "molecular" exchange field is instrumental for the so-called JP mechanism[57–59] that could account for the reentrant hfSC phase.

## Jaccarino-Peter compensation effect in UTe$_2$

Before we discuss this JP mechanism, let us summarize briefly the present situation for the various superconducting phases in UTe$_2$ at ambient pressure. In zero field, there is a consensus for UTe$_2$ being recognized as a candidate spin triplet superconductor with a B$_{3u}$ or A$_u$ symmetry. Finer details such as the nodal structure are still under debate[39]. Recently, several experiments have revealed a clear phase transition to another superconducting phase for field along the $b$ axis above ≈15 T at low temperatures[19,60,61]. Theoretical proposals anticipated a transition to a B$_{2u}$ symmetry ($d$ vector with no component along the $b$ axis). In contrast, thermodynamic experiments have revealed drastic changes between the two phases, suggesting a change of the pairing mechanism in addition to a symmetry change[19]. It has also been shown that a spin-singlet state can account for the observed strong broadening of the superconducting anomaly as well as its angular dependence[19]. Regarding the hfSC phase, as opposed to the phases below $H_m$, there are no theoretical models yet, and the common wisdom is that it should be spin-triplet to survive such high fields. The initial proposed mechanisms were a JP or a Lebed mechanism[12], dismissed or abandoned for that of a "Landau level superconductivity"[32,62]. This last hypothesis is rather surprising when contrasted with the existence of the hfSC phase even in very dirty systems[62].

In the following, we propose a JP scenario mainly relying on a spin-singlet state. As shown already for the field-reinforced superconducting phase along the $b$ axis, spin-singlet SC is able to survive in such high fields supported by strong-coupling effects and the field-induced reinforcement of the pairing strength[19]. At the root of the JP mechanism is a compensation between the external field, $H$, and an internal exchange field, $H_{ex}$[57]. The latter is associated with the polarization of local magnetic ions and acts on the spin of the itinerant quasiparticles. Compensation is possible only if $H_{ex}$ is opposite to $H$, which requires an antiferromagnetic exchange coupling between local and itinerant spins. In the case of UTe$_2$, the local moments originate from the uranium ions. At the mean-field level, $H_{ex}$ can be expressed as: $H_{ex} = J_c\langle M_c\rangle\hat{c} + J_b\langle M_b\rangle\hat{b}$, with $J_c$ and $J_b$ and $\langle M_c\rangle$ and $\langle M_b\rangle$, respectively, the anisotropic exchange constants and magnetization components along the $c$ and $b$ axes. Hence, at finite tilts within the $(b, c)$ plane, the direction of $H_{ex}$ is most likely not perfectly collinear with $H$. Nevertheless, antiferromagnetic coupling (negative $J_c$ and $J_b$) is quite natural for such a Kondo-lattice system. If $H$ and $H_{ex}$ compensate each other, then the itinerant quasiparticles feel no Zeeman field and they should lose their polarization: Our Hall-effect results suggest that in the angular range around 30°, the compensation between both fields is quite efficient, at least around 70 T. The marked decrease of the torque at 25° and low temperatures, mentioned earlier, also indicates that for this angle, the magnetization above $H_m$ is closer to the field direction. This is beneficial for the compensation of $H_{ex}$ by $H$ in neighboring angles (once taking into account exchange anisotropy). We see two possible ways for how the hfSC phase arises via this JP compensation mechanism.

In the first scenario, the superconducting pairing is restored by an absence of band polarization (and at the opposite, suppressed when the magnetization is saturated). This would work both for a spin-singlet and a spin-triplet superconducting order parameter.

In the second scenario, $H_{ex}$ directly counters the paramagnetic limit enabling the restoration of SC: this would be a "true" JP compensation[57,63,64], requiring that the hfSC phase is spin singlet or

spin triplet with a sizeable $d$ component along the applied field: $H_{ex}$ solely acts on the spins, i.e., when the compensation of $H$ and $H_{ex}$ becomes perfect at a certain field, $H_{c2}$ remains restricted only by the orbital limit. Stunning examples were found among organic superconductors with field-reentrant phases attributed to the JP compensation effect[65,66]. However, the observed angular dependencies were extremely sensitive to the field alignment due the huge anisotropy of the orbital limit in these 2D materials. In both proposed scenarios, a natural assumption is that the reentrant hfSC phase is a resurgence of the field-reinforced superconducting phase observed for a narrow angular range (few degrees) about $H \parallel b$, persistent up to $H_m$. Otherwise, yet another pairing mechanism should take place in UTe$_2$ above $H_m$.

We have discussed already (see Fig. 2e) how the hfSC phase could exist for $H_{c2}$ only limited by the orbital effect, thanks to an increase of $T_c$ of up to about 3 K without any change of the bare $v_F$ as compared to the lfSC phase. This hypothesis of pure orbital limiting requires an ESP spin-triplet state for the hfSC phase. Hence, the compensation effect could only act on the value of $\lambda$ (first scenario). The results of Fig. 2e show that in such a case, $\lambda$ (see dashed lines in Fig. 4b) would be essentially field independent above $H_m$ with a value changing only little (between 1.5 and 1.6) within the angular range of the hfSC phase. As a consequence, $\lambda$ above $H_m$ should grow with the tilt angle. This, however, stands in contrast to the quick vanishing of the field-reinforced SC beyond only few degrees tilt. Moreover, for the JP compensation to work effectively, $\lambda$ at finite angle should be at most of the order of that along the $b$ axis in the field-reinforced phase just below $H_m$.

By contrast, the second scenario involving a true JP effect does not suffer from these caveats, if we assume a spin-singlet phase below $H_m$ for $H \parallel b$[19]. The distinction to the first scenario is that in this case below $H_m$, SC is mainly controlled by the Pauli depairing. Hence, the fast suppression of SC for only small tilts away from $H \parallel b$ can be attributed to the lowering of the Pauli limit at constant field, when $\lambda(H)$ decreases due to the increase of $H_m$ with angle (see Supplementary Note 5). Then, even partial compensation of this paramagnetic limit by $H_{ex}$ above $H_m$ at finite angle can restore SC without requiring to surpass $\lambda$ for $H \parallel b$.

In order to obtain a proper model that can describe the location of the hfSC pocket in the ($\theta$, $T$, $H$) phase space, and notably the $T$ dependence of $H_{c2}$ at a given $\theta$ in the JP scenario, we need to know the $H$ dependence of $\lambda$ together with the degree of compensation of the paramagnetic limit. Presently, too little is known about the magnetization and $H_{ex}$ above $H_m$ (see discussion in Supplementary Note 5). Thus, there are (too) many possible tuning parameters for a comprehensive quantitative model of the hfSC pocket. Nevertheless, we can attempt a modeling of our angle-dependent $H_{c2}$ results. In order to fix the $H$-dependent compensation of the paramagnetic limit controlled by $g\mu_B(H - H_{ex})$, we use $H_{ex} = 70$ T under the assumption that it remains constant and parallel to $H$. Hence, $\lambda(H)$ can be extracted from the data of Fig. 2e. The result is shown in Fig. 4b. We find that the JP compensation scenario leads to a decrease of $\lambda$ just above $H_m$, diminishing further for $H > H_m$. Again, the key point here is the dominant role of the paramagnetic limit, i.e., this controls the disappearance of SC at finite angle below $H_m$ and its reentrance above $H_m$ due to the compensation. More details on the model are given in Supplementary Note 5. In addition, the proposed mechanism can also explain the recently reported pressure dependence of the hfSC phase[32]. Under pressure, SC was found to survive at finite angles up to $H_m$, or even to exist above $H_m$ detached from the metamagnetic transition line.

The mechanism behind the hfSC phase and its relation to the SC at lower fields is under hot debate[12,62] and still without even a qualitative satisfying scenario. Here, we show how the JP compensation effect, dismissed by previous studies, can explain the hfSC phase. This is supported by the vanishing of the Hall effect. This scenario of spin-singlet SC in the hfSC phase is also connected to the same state

proposed for the field-reinforced superconducting phase emerging below the metamagnetic transition for $H \parallel b$[19].

In summary, our study features insights on the enigmatic high-field properties of the putative spin-triplet heavy-fermion superconductor UTe$_2$. We demonstrate by torque magnetometry that the magnetization jump at the metamagnetic transition and the applied field are noncollinear, keeping a large component along the $b$ axis. This is probably related to the observed $1/\cos\theta$ dependence of the metamagnetic transition field, $H_m$. We studied angle-dependent magnetotransport in 70 T pulsed magnetic fields. Here, we focused particularly on the distinct high-field superconducting phase induced just above $H_m$ for tilt angles of around 35° within the ($b$, $c$) plane surviving very high field values above 40 T. We have determined the angular dependence of the upper critical field, $H_{c2}$, in this phase, reaching a maximum of $\mu_0 H_{c2} \approx 73$ T. This value is amongst the highest reported for heavy-fermion superconductors. Our studies reveal an apparent correlation between $H_{c2}$ and the normal-state Hall effect at very high fields. The latter exhibits a minimum with an almost complete suppression, precisely where the reentrant hfSC emerges and reaches its maximum robustness. The analyses of this correlation hints at a compensation mechanism as the potential origin of both phenomena: In the angular region around 35°, compensation between the exchange field above $H_m$ and the applied field, such that band polarization is strongly reduced, is consistent with our observations. A reduced band polarization can lead to the suppression of the dominant AHE contribution and to a reentrant superconducting phase (JP effect). Our results provide a guide for future experiments and theory that will show more quantitatively if and how this may appear. Such a scenario puts specific constraints on the potential order parameter of the superconducting phase discussed in our work. Solving the riddle of how Cooper pairs, built by heavy quasiparticles, can survive in extreme magnetic fields will certainly help advance our fundamental understanding of unconventional superconductors.

## Methods

### Crystal growth

The UTe$_2$ single crystals were prepared as described in ref. 7. All single crystals were prepared by the chemical vapor transport method with iodine as transport medium. A starting ratio of U:Te = 2:3 has been used, and the quartz ampules were heated slowly up to a final temperature of 1000 °C on one side and 1060 °C on the other side and this temperature gradient was maintained for 18 days. The ampules were slowly cooled down to ambient temperature during 70 h.

### Microcantilever torque magnetometry

For magnetic-torque experiments, we cut samples with dimensions $(100 \times 20 \times 3)$ $\mu m^3$ from a single crystal using FIB assisted etching. We used a Wheatstone-bridge-balanced piezo-resistive cantilever (eigenfrequency ~300 kHz)[33]. The sample was attached by Apiezon (N) grease. The setup was mounted on a rotator, such that the angle between field and cantilever could be varied, and installed in a $^3$He cryostat. Pulsed magnetic fields of up to 70 T were applied. An example picture of the microcantilever including a sample attached to it is presented in the inset of Fig. 1a.

### FIB-microfabrication of transport devices

Device #1, shown in Fig. 1b, was fabricated in the following steps: First, a slice $(150 \times 20 \times 2)$ $\mu m^3$ was separated out of the crystal using FIB and transferred ex situ onto a sapphire chip. Next, an approximately 150 nm thick layer of gold was sputter deposited covering a rectangular area around and including the sample slice. In a next step, carbon-rich platinum was deposited in a FIB system at the two ends and at six side points around the sample slice (see Fig. 1b). The platinum fixations establish a galvanic connection between the gold layer on the chip and the top surface of the sample. Next, the gold layer was partially etched

away from the central top surface of the sample by ions. Then, a focused ion beam was applied to cut trenches into the gold layer and the sample in order to create well-defined terminals. Resistances of a few ohms were achieved. In the end, a droplet of transparent unfilled Stycast hardened in vacuum was used to protect the structured device from air. We fabricated three different devices for this study with the following width, thickness, and length ($w \times d \times l$) between the contacts: device #1 $(10 \times 2.7 \times 75)\,\mu m^3$; device #2 $(4 \times 2.9 \times 58)\,\mu m^3$; device #3 $(7.1 \times 4.5 \times 48.5)\,\mu m^3$.

## Magnetotransport measurements

We performed steady-field characterization measurements in an Oxford dilution refrigerator equipped with an 18 T superconducting magnet. We measured the resistance with a standard a.c. four-point lock-in technique. We conducted pulsed high-field experiments at the Dresden High Magnetic Field Laboratory in a 60 T and 70 T pulsed-magnet systems with a pulse duration of 25 ms and 150 ms, respectively, equipped with either $^4$He and $^3$He cryostat inserts.

## Data availability

All data that support the findings of this study are available from the corresponding author upon request.

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

## Acknowledgements
We thank J. Flouquet, H. Suderow, A. Miyake, Y. Yanase, M. Zhitomirsky and J. Gayles for fruitful discussions. The help by N. Nagaosa in the analysis of the anomalous Hall effect was particularly enlightening. We thank A. Mackenzie for continuous support. We thank S. Seifert for his technical assistance. D.A. acknowledges support by KAKENHI(JP19K03736, JP19H00646, JP20K20889, JP20H00130, JP20KK0061, JP22H04933), ICC-IMR. We acknowlege the support from GP-spin at Tohoku University and JSPS. A.P., G.L., G.K. and J.P.B. acknowledge support from CEA Exploratory program TOPOHALL, the French National Agency for Research ANR within the project FRESCO No. ANR-20-CE30-0020, FET-TOM No. ANR-19- CE30-0037 and SCATE ANR-22-CE30-0040. T.H. acknowledges support from the German Research Foundation (DFG) Grant No. HE 8556/3-1. We acknowledge the support of the HLD at HZDR, member of the European Magnetic Field Laboratory (EMFL), and the DFG through the Würzburg-Dresden Cluster of Excellence on Complexity and Topology in Quantum Matter - ct.qmat (EXC 2147, project.id 390858490).

## Author contributions
T.H., K.S., M.Ki. and M.Kö. were involved in the preparation of the devices by FIB microfabrication. T.H., K.S., A.M., T.F. and M.Ki. performed the pulsed-field experiments. T.F. and T.H. performed high-field torque magnetometry. J.H., J.S. and T.H. performed low-field characterizations. G.L., D.A. and G.K. prepared the high-quality samples. J.-P.B., T.H., K.S. and A.P. analyzed the upper critical field and the Hall-effect data. T.H., M.Ki., K.S., T.F., J.H., M.Kö., I.S., A.P., D.A., G.K., J.W. and J.-P.B. took part in discussing the results and writing the manuscript.

## Funding

## Competing interests
The authors declare no competing interests.
