## [Peer Review File · Nature Communications]

Field-induced compensation of magnetic exchange as the possible origin of reentrant superconductivity in UTe₂Editorial Note: Parts of this Peer Review File have been redacted as indicated to remove third-party material where no permission to publish could be obtained.

REVIEWER COMMENTS

Reviewer #1 (Remarks to the Author):

The manuscript "Suppressed magnetic scattering sets conditions for the emergence of 40 T high-field superconductivity in UTe₂" presents pulsed-field studies of UTe₂, extrapolating an extremely high upper critical field of 75 T for the high field superconducting phase that was previously reported in this compound. The authors establish a high-field phase diagram based on electrical transport and magnetization data, and attempt to fit the high-field boundary of the superconducting phase with a phenomenological model. While a comprehensive theory to describe this highly unusual superconducting phase is still lacking, this work lays the foundation for future theoretical studies of the compound, providing high-quality experimental data in the challenging environment of extreme magnetic fields. This paper provides new and unique information on UTe₂, a compound of great current interest, and it is well written and convincingly argued. I therefore recommend publication in Nature Communications, with a few further comments and suggestions below.

1) The authors fit the phase boundary of the high field superconducting phase assuming a constant coupling constant, to determine an upper limit for the critical field, and they estimate the zero-field critical temperature from this critical field. The characteristic energy for the superconducting pair formation would have to be increased to support this higher T_c, which would be important evidence of a different pairing mechanisms. It is also mentioned that the characteristic energy is in good agreement with energy scales observed for magnetic fluctuations in neutron scattering. However, the authors later argue that the increase in T_c could also be due to a field-dependent increase in the strong coupling parameter. In this case, it would be helpful if the authors commented on why they chose to analyze the data with a fixed coupling constant, or to add a comparative analysis where the coupling constant is allowed to vary with the magnetic field.

2) In addition to electrical resistivity data, the authors present measurements of the Hall effect as evidence for a superconducting state (vs a possible low-resistance state). The observed zero Hall effect indeed supports a superconducting state. Considering the possible topological nature of UTe₂, an alternative scenario could be envisioned, where the zero Hall effect is a consequence of transport in topologically protected surface states. It may be interesting to consider future experiments distinguishing bulk and surface transport, for example in the Corbino geometry.

3) The authors do not mention the sample quality, other than it being FIB-cut pieces of a single crystal. The critical temperature is rather typical for the compound, however it would be helpful to compare the sample quality in this work to other samples of UTe₂ with reported critical temperatures around 2 K, if the data exists.

4) Language: while the paper is well written overall, there are still a few typos and incomplete sentences that should be corrected, e. g. :

- page 5, second sentence: "Thanks to the sample preparation by FIB the volume of the sample is small enough to prevent the microcantilever from [missing text here]".

- page 9, bottom: "Such a behavior would be the symmetric of that observed for [...]" -> intended meaning: "Such a behavior would be similar to that observed for [...]"

- page 11, subheading: "Strong suppression of the anomalous Hall in the vicinity of the hfSC phase" -> change to "Strong suppression of the [...] Hall Effect [...]"

- page 13, top: "provides further proof of the gapping out of charge carriers" -> replace "gapping" with "gapping"

- page 13, second paragraph: "[...] it is tempting to suggest that the emergence of hfSC at tilted field orientation may be correlated with the change in the influence of the [missing text here]".

I am grateful to have been given the opportunity to review this manuscript, and commend the authors on their impressive experimental results!

Sincerely,
Lucia Steinke

Reviewer #2 (Remarks to the Author):

please see the attached file.

UTe₂ exhibits a specific high field induced superconductivity when magnetic field is applied about 30 degrees between b axis and c axis. This high field SC exists in a very large field range, and therefore attracted a lot of research interest. The authors investigated this phase using magnetic torque and electric transport measurements on devices fabricated by FIB. There are many good results from this study, which should certainly be published in some format. However, I am not sure whether it is a good fit to Nature Communications.

Here is my biggest concern. The paper is not written around one central topic. Instead, it is consisted of a few related parts. Each part has some good results, but not very significant, as shown below:

1. It used magnetic torque to probe the metamagnetic transition as a function of angle. It is a good complementary to the magnetization data. But new knowledge obtained from these measurements are limited. Basically it confirms that there is large anisotropy, which people already know.
2. The resistance data on FIB devices confirmed the data on bulk sample for hfSC phase. It is good to know that FIB devices, with better geometry, can repeat the same results. Again, the new knowledge obtained is very limited.
3. H_{c2} of hfSC phase is fitted and H_{c2} is determined to be around 75 T. Previous studies show that H_{c2} is beyond 65 T, but not much beyond. Now it is determined to be 75 T with larger field and fitting. It is certainly an improvement, but I guess it is not that important for broad audience. As they mentioned, there is not theoretical model of hfSC phase. Their fitting contains a lot of assumptions, and the results show T_c of 3K which is contradict to the observed value of 1.6 K. I don't want to complain about their fitting, as it is a good try without an existing model. But I don't want to take it too seriously, as the results obviously go against the experiments.
4. Hall data shows zero signal for the hfSC phase. It confirms that hfSC phase is SC phase. There are many arguments to believe hfSC phase is SC phase (e.g., arguments in arXiv:2206.06508). Of course, it is still very good to show zero Hall signal. However, the data is not that definitive either. For angle around 30, R is not zero between 15 and 45T, not SC, therefore Hall signal should not be zero in this field range either. Figure 4b shows "zero" Hall signal for 30 degree up to 55T. This indicates that the measurement does not have enough resolution to tell the small non zero Hall (between 15 and 45T) and zero Hall (below 15T and between 45 and 55T). Therefore the claim of zero Hall in hfSC (even though I believe it must be zero) is not definitive based on current data set.
5. The title of the paper is not discussed until the last part. It shows the slope of the Hall data in large field is suppressed around 30 degree. This is the only new information of the paper, and I guess therefore is chosen as the title. However, the results was only mentioned in one paragraph and discussed in another paragraph. The short discussion of these data reflects that fact that these data can not be well explained without further experiments. As matter of fact, the discussion part of these data is mainly to say more data, such as magnetization, is needed to really understand the Hall. Basically, there is new data about hfSC phase, but we don't know what it means.

To summarize, the manuscript presents a few good results, each representing a small progress to understanding the hfSC phase of UTe₂. Each alone is not very a significant contribution. Does

combining many small progresses make the overall paper significant? I'll leave that judgement to the editor.

Besides this major concern, I have to point out there are many issues related to the writing:

1. First sentence of 2nd paragraph on page 5, "sample is small enough to prevent the microcantilever from". Clearly something is missing after from.
2. Last paragraph on page 13, "it is tempting to suggest that the emergence of hfSC at tilted field orientation may be correlated with the change in the influence of the". Clearly something is missing after the.
3. In first sentence on page 5, "its the response" the is a typo.
4. 1st paragraph on page 7, "Similarly, to what we observed by magnetic torque the metamagnetic transition shifts towards higher fields." The grammar is not correct.
5. In the first paragraph of the introduction, there is a sentence: "Profound experimental evidence in UTe₂ has recently been provided by NMR, STM and polar Kerr effect studies." Evidence for what? It is not clear from the context.
6. All of sudden, it appears a very short paragraph, 3rd of introduction, talking about the FM fluctuations. 2nd paragraph talks about the high field SC phase, and 4th paragraph talks about mysteries of the mechanism. These two paragraphs naturally go together. Now it is cut by a wired 3rd paragraph. Although FM fluctuation is certainly a related topic, it should find a better place.
7. The main point of the 4th paragraph is that most quantities do not change as function of angle near 30 degree, therefore HfSC is not well understood. At the end of the paragraph, it appears a sentence talking about the field reinforced SC along b axis is due to the enhancement of magnetic fluctuations. There is no logic continuity between the last sentence and the rest paragraph.
8. The last paragraph on page 3 first talks about the order parameter. For example, whether the order parameter has multicomponent. Then it says to understand the mechanism of how hfSC is induced is the key question. Again, there is no logic continuity.
9. Page 5, 3rd paragraph, "Upon decreasing temperature, the metamagnetic transition evolves from a broad second-order into a sharp first-order type transition." I assume this refers to the resistance data. Comparing figure 1e and 1f, it is not clear which one has broad transitions and which one has sharp ones. Can authors clarify?
10. Last paragraph on page 11, "Previous pulsed-field studies of resistivity and magnetization have already reported a zero-resistance state including a drop in the magnetization indicating the hfSC phase". Ref. 10 did not show drop in magnetization. Drop in magnetization has not been shown as far as I know, including the magnetic torque data in the current study. "a low non-zero resistivity in combination with a drop in magnetization may indicate a very metallic state". Even though ref. 10 showed low non-zero resistance, ref. 15 did show zero resistance data.

Reviewer #3 (Remarks to the Author):

The authors present a high magnetic field study of the putative spin triplet superconductor UTe₂. The novelty in their approach is to use samples that have been machined using FIB techniques to control sample dimensions.

Furthermore, they present Hall effect data at high fields.

The authors reproduce measurements to show that their sample preparation technique does not adversely affect the sample properties. They present an incremental step in measured properties in UTe₂.

The quality of the samples is not well described. Recently, samples with higher T_c have been synthesized, and therefore, the results presented may be from sub-optimally grown crystal.

The authors claim that the Hall resistance shows negative derivative in the field region of 65 to 69T. However, this is a rather small field range, and it is not clear if the negative slope in the superconducting region (with rather large noise) is taken into account.

The large experimental space, magnetoresistance at different temperatures, magnetic fields, and different tilt angles away from the b-axis present is not well discussed, and limits the readership.

Response to reviewer comments

We are grateful for this opportunity to defend our work and resubmit a revised version. We thank the reviewers for their time and constructive criticism. Our manuscript underwent a substantial revision. Many of the mistakes and issues mentioned in the reports were removed in this process. We hereby provide a point-by-point response to the reviewer comments.

Reviewer #1 (Remarks to the Author):

The manuscript "Suppressed magnetic scattering sets conditions for the emergence of 40 T high-field superconductivity in UTe₂" presents pulsed-field studies of UTe₂, extrapolating an extremely high upper critical field of 75 T for the high field superconducting phase that was previously reported in this compound. The authors establish a high-field phase diagram based on electrical transport and magnetization data, and attempt to fit the high-field boundary of the superconducting phase with a phenomenological model. While a comprehensive theory to describe this highly unusual superconducting phase is still lacking, this work lays the foundation for future theoretical studies of the compound, providing high-quality experimental data in the challenging environment of extreme magnetic fields. This paper provides new and unique information on UTe₂, a compound of great current interest, and it is well written and convincingly argued. I therefore recommend publication in Nature Communications, with a few further comments and suggestions below.

1) The authors fit the phase boundary of the high field superconducting phase assuming a constant coupling constant, to determine an upper limit for the critical field, and they estimate the zero-field critical temperature from this critical field. The characteristic energy for the superconducting pair formation would have to be increased to support this higher T_c, which would be important evidence of a different pairing mechanisms. It is also mentioned that the characteristic energy is in good agreement with energy scales observed for magnetic fluctuations in neutron scattering. However, the authors later argue that the increase in T_c could also be due to a field-dependent increase in the strong coupling parameter. In this case, it would be helpful if the authors commented on why they chose to analyze the data with a fixed coupling constant, or to add a comparative analysis where the coupling constant is allowed to vary with the magnetic field.

This is a valid point. The simple answer is that at that time we did not really had a model for the mechanism behind the field dependence. For such a fit, a model is required that suggests the relation between the coupling strength and the external field. Thanks to more recent findings by some of the coauthors [Rosuel et al. PRX 13, 011022 (2023), Ref. 14] a field enhancement of the distinct SC emerging close to the metamagnetic transition can be expected. In our revised manuscript, we took that model into account. We now also discuss the scenario of the Jaccarino-Peter compensation as the potential mechanism behind the high-field superconducting phase. It may be the driving force behind the reentrance of SC above H_m at tilt angles explaining our observations in transport. As we now discuss, there is strong evidence from the steep suppression of the Hall component (presumably an anomalous Hall effect of intrinsic origin) upon field rotation. In the new discussion as well as in Supplementary Note 5 we now provide an attempt to fit our H_{c2}(T) data under the assumption of a Jaccarino-Peter compensation. Fig. 4 and Fig. S5 demonstrate the field dependence of the coupling strength and the feasibility of the fit. Nevertheless, with the

large number of parameters that need to be taken into account we cannot provide an unambiguous model at present. We therefore, try to make sensible conclusions that may guide further experiments and theoretical studies.

2) In addition to electrical resistivity data, the authors present measurements of the Hall effect as evidence for a superconducting state (vs a possible low-resistance state). The observed zero Hall effect indeed supports a superconducting state. Considering the possible topological nature of UTe₂, an alternative scenario could be envisioned, where the zero Hall effect is a consequence of transport in topologically protected surface states. It may be interesting to consider future experiments distinguishing bulk and surface transport, for example in the Corbino geometry.

The suggested experiment seems an exciting approach to the question of the topological properties of UTe₂. It would be interesting to see if such a measurement configuration would yield any insights. However, owing to the large current used in the experiments, it is unlikely that we could already be sensitive to such surface states. Note also that they remain largely speculative in UTe₂.

3) The authors do not mention the sample quality, other than it being FIB-cut pieces of a single crystal. The critical temperature is rather typical for the compound, however it would be helpful to compare the sample quality in this work to other samples of UTe₂ with reported critical temperatures around 2 K, if the data exists.

This sample is already much better than the ones used in the first experiments reported by Ran et al. [Science 2019]. For the 2 K samples, we can expect an even larger pocket of high field superconductivity, but this does not change its existence for these samples.

Furthermore, in the mentioned study [Rosuel et al. PRX 13, 011022 (2023), Ref. 14] a comparison of thermodynamic probes for multiple sample with varying T_c is presented. There is no apparent difference in the fundamental physics for these samples. We, therefore, are confident about the relevance of our study.

4) Language: while the paper is well written overall, there are still a few typos and incomplete sentences that should be corrected, e. g. :

- page 5, second sentence: "Thanks to the sample preparation by FIB the volume of the sample is small enough to prevent the microcantilever from [missing text here]".

We added the missing content.

- page 9, bottom: "Such a behavior would be the symmetric of that observed for [...]" -> intended meaning: "Such a behavior would be similar to that observed for [...]"

We revised that sentence.

- page 11, subheading: "Strong suppression of the anomalous Hall in the vicinity of the hfSC phase" -> change to "Strong suppression of the [...] Hall Effect [...]"

We modified the section header.

- page 13, top: "provides further proof of the gapping out of charge carriers" -> replace "gapping" with "gapping"

We corrected that typo.

- page 13, second paragraph: "[...] it is tempting to suggest that the emergence of hfSC at tilted field orientation may be correlated with the change in the influence of the [missing text here]".

We added the missing content.

I am grateful to have been given the opportunity to review this manuscript, and commend the authors on their impressive experimental results!

We thank the reviewer for her overall positive feedback and constructive comments.

Reviewer #2 (Remarks to the Author):

UTe₂ exhibits a specific high field induced superconductivity when magnetic field is applied about

30 degrees between b axis and c axis. This high field SC exists in a very large field range, and therefore attracted a lot of research interest. The authors investigated this phase using magnetic torque and electric transport measurements on devices fabricated by FIB. There are many good results from this study, which should certainly be published in some format. However, I am not sure whether it is a good fit to Nature Communications.

Here is my biggest concern. The paper is not written around one central topic. Instead, it is consisted of a few related parts. Each part has some good results, but not very significant, as shown below:

1. It used magnetic torque to probe the metamagnetic transition as a function of angle. It is a good complementary to the magnetization data. But new knowledge obtained from these measurements are limited. Basically it confirms that there is large anisotropy, which people already know.

Our results from magnetic torque provide further confirmation of the overall angle dependence of the transition occurring at H_m . They clearly demonstrate that the overall angle dependence of the magnetization is monotonous and does not exhibit any steep/complete suppression as observed in the Hall response. Moreover, the previous magnetization experiments [Miyake et al. JPSJ 90 (2021), Ref. 23, Ran et al. Nat. Phys. 15 (2019), Ref. 7] were only sensitive to the component parallel to the field. Our torque results demonstrate that there appears to be a significant perpendicular component even above the metamagnetic transition in the high-field phase. This is an important new information, better put forward in the revised manuscript, which is essential for a proper (future) modeling of the transition at H_m , and for that of the compensation mechanism. It also plays an important role in the analysis of the anomalous Hall effect (AHE), showing that the stunning suppression of the AHE around 30° is not originating from any anomaly in the magnetization nor in the resistivity. The origin of the effect has deeper grounds, and we propose now that it marks a drastic reduction of the band polarization in this angular range.

2. The resistance data on FIB devices confirmed the data on bulk sample for hfSC phase. It is good to know that FIB devices, with better geometry, can repeat the same results. Again, the new knowledge obtained is very limited.

See our reply above. It is actually striking that both magnetization and magnetoresistance do not exhibit any significant angle-dependent features. This provides evidence for an enhanced intrinsic Hall component for $H \parallel b$ that seems to get suppressed around $\theta = 30^\circ$.

3. H_{c2} of hfSC phase is fitted and H_{c2} is determined to be around 75 T. Previous studies show that H_{c2} is beyond 65 T, but not much beyond. Now it is determined to be 75 T with larger field and fitting. It is certainly an improvement, but I guess it is not that important for broad audience.

We actually disagree. Our results and in particular the high H_{c2} and the explanation by the Jaccarino-Peter compensation demonstrate to a broader readership the exciting features

(yet with the potential to expand and enhance even further) observable in the field of heavy-fermion compounds. In particular, the enhancement of H_{c2} and potential mechanisms that may enable this are interesting from a theoretical, materials and application point of view.

As they mentioned, there is not theoretical model of hfSC phase. Their fitting contains a lot of assumptions, and the results show T_c of 3K which is contradict to the observed value of 1.6 K. I don't want to complain about their fitting, as it is a good try without an existing model. But I don't want to take it too seriously, as the results obviously go against the experiments.

We would kindly express our disagreement with this conclusion. We have shown that, T_c (and, hence, λ) needs to experience an increase with field in order to account for such high H_{c2} values. Our fits reproduce the overall field dependence even with the rather simple assumptions. There is no contradiction with the experiments. We show the fits in an extended representation in order to highlight the necessary enhancement of T_c . However, from the experiments it already becomes apparent that there is an additional mechanism, likely linked to H_m , that quenches the lfSC at fields below H_m . This point was not captured in the presentation. In our revised manuscript, however, we now provide a first suggestion for a model based on an exchange-compensation effect driven by magnetic field and referred to as the Jaccarino-Peter effect. As we now show, this hypothesis matches our observation for the anomalous Hall effect and can help in understanding the overall angle-dependent phase diagram.

4. Hall data shows zero signal for the hfSC phase. It confirms that hfSC phase is SC phase. There are many arguments to believe hfSC phase is SC phase (e.g., arguments in arXiv:2206.06508). Of course, it is still very good to show zero Hall signal. However, the data is not that definitive either. For angle around 30, R is not zero between 15 and 45T, not SC, therefore Hall signal should not be zero in this field range either. Figure 4b shows “zero” Hall signal for 30 degree up to 55T. This indicates that the measurement does not have enough resolution to tell the small non zero Hall (between 15 and 45T) and zero Hall (below 15T and between 45 and 55T). Therefore the claim of zero Hall in hfSC (even though I believe it must be zero) is not definitive based on current data set.

We agree that our presentation may have been a bit misleading. We would like to stress that our high-resolution measurement with relatively high current (500 μ A) shows a clear zero in the high-field range. The data that the reviewer is referring to was recorded for a 5 times lower current. The noise level therefore was simply higher. However, we show that in the normal state the up- and down-sweep data coincide much better than in the case of the larger current settings. Hence, the Hall effect results can be analyzed with more confidence than for the low-resolution (low current) data set.

5. The title of the paper is not discussed until the last part. It shows the slope of the Hall data in large field is suppressed around 30 degree. This is the only new information of the paper, and I guess therefore is chosen as the title. However, the results was only mentioned in one paragraph and discussed in another paragraph. The short discussion of these data reflects that fact that these data can not be well explained without further experiments. As matter of fact, the discussion part of these data is mainly to say more data, such as magnetization, is needed to really understand the Hall. Basically, there is new data about hfSC phase, but we don't know what it means.

To summarize, the manuscript presents a few good results, each representing a small progress to understanding the hfSC phase of UTe₂. Each alone is not very a significant contribution. Does

combining many small progresses make the overall paper significant? I'll leave that judgement to the editor.

We hope the comments above have already explained the importance of the presented results. In our revised manuscript, we now provide a first hypothesis based on these results that can explain the high-field phase diagram. Most importantly, in order to support the idea of the Jaccarino-Peter compensation mechanism, the data reported herein needs to be taken into account in its entirety. The absence of major features in magnetoresistivity and torque upon variation of the field orientation, the enhancement of the metamagnetic transition field with angle, the angle dependence of H_{c2} and the suppression of the anomalous Hall component support our conclusion. Our observations surely will have a significant impact on the overall understanding of UTe_2 .

Besides this major concern, I have to point out there are many issues related to the writing:

1. First sentence of 2nd paragraph on page 5, “sample is small enough to prevent the microcantilever from”. Clearly something is missing after from.

We corrected these typos and made sure that the overall presentation is coherent.

2. Last paragraph on page 13, “it is tempting to suggest that the emergence of hfSC at tilted field orientation may be correlated with the change in the influence of the”. Clearly something is missing after the.

Corrected.

3. In first sentence on page 5, “its the response” the is a typo.

Corrected.

4. 1st paragraph on page 7, “Similarly, to what we observed by magnetic torque the metamagnetic transition shifts towards higher fields.” The grammar is not correct.

Corrected.

5. In the first paragraph of the introduction, there is a sentence: “Profound experimental evidence in UTe_2 has recently been provided by NMR, STM and polar Kerr effect studies.” Evidence for what? It is not clear from the context.

Evidence for topological character of the SC ground state`

We revised the introduction and considered this comment.

6. All of sudden, it appears a very short paragraph, 3rd of introduction, talking about the FM fluctuations. 2nd paragraph talks about the high field SC phase, and 4th paragraph talks about mysteries of the mechanism. These two paragraphs naturally go together. Now it is cut by a wired 3rd paragraph. Although FM fluctuation is certainly a related topic, it should find a better place.

We revised the introduction and focused its content a bit more towards our main findings.

7. The main point of the 4th paragraph is that most quantities do not change as function of angle near 30 degree, therefore HfSC is not well understood. At the end of the paragraph, it appears a sentence talking about the field reinforced SC along b axis is due to the enhancement of magnetic fluctuations. There is no logic continuity between the last sentence and the rest paragraph.

This has been revised into a more cohesive presentation.

8. The last paragraph on page 3 first talks about the order parameter. For example, whether the order parameter has multicomponent. Then it says to understand the mechanism of

how hfSC is induced is the key question. Again, there is no logic continuity.

9. Page 5, 3rd paragraph, “Upon decreasing temperature, the metamagnetic transition evolves from a broad second-order into a sharp first-order type transition.” I assume this refers to the resistance data. Comparing figure 1e and 1f, it is not clear which one has broad transitions and which one has sharp ones. Can authors clarify?

10. Last paragraph on page 11, “Previous pulsed-field studies of resistivity and magnetization have already reported a zero-resistance state including a drop in the magnetization indicating the hfSC phase”. Ref. 10 did not show drop in magnetization. Drop in magnetization has not been shown as far as I know, including the magnetic torque data in the current study. “a low non-zero resistivity in combination with a drop in magnetization may indicate a very metallic state”. Even though ref. 10 showed low non-zero resistance, ref. 15 did show zero resistance data.

Thank you for these corrections and suggestions. We have considered them in our revised version.

Reviewer #3 (Remarks to the Author):

The authors present a high magnetic field study of the putative spin triplet superconductor UTe₂. The novelty in their approach is to use samples that have been machined using FIB techniques to control sample dimensions.

Furthermore, they present Hall effect data at high fields.

The authors reproduce measurements to show that their sample preparation technique does not adversely affect the sample properties. They present an incremental step in measured properties in UTe₂.

The quality of the samples is not well described. Recently, samples with higher T_c have been synthesized, and therefore, the results presented may be from sub-optimally grown crystal.

This concern is similar to point 3 raised by the first reviewer: we can repeat here our answer: The investigated samples are already much better than the ones used in the first experiments reported by Ran et al. [Science 365, 684 (2019), Ref. 1], and with little difference. For the 2 K samples, we can expect an even larger pocket of high field superconductivity, but this does not change its existence for these samples. Furthermore, in the recent study by some of the coauthors [Rosuel et al. PRX 13, 011022 (2023), Ref. 14] a comparison of thermodynamic probes for multiple sample with varying T_c is presented. There is no apparent difference in the fundamental physics for these samples. Our sample originates from the same origin. We therefore are confident about the quality of our study.

The authors claim that the Hall resistance shows negative derivative in the field region of 65 to 69T. However, this is a rather small field range, and it is not clear if the negative slope in the superconducting region (with rather large noise) is taken into account.

In the revised manuscript, we have re-analyzed our data and completely changed the presentation and discussion on the Hall effect. Even though the negative slope clearly emerges from the noise level, it is not the main point, as the Hall coefficient is not the derivative of ρ_{xy} but the ratio ρ_{xy}/H . Hence, the key feature we emphasize in this new version is the fact that the overall Hall signal (not the derivative) is suppressed at around 30°. This is a robust feature shining a new light onto the special angular region around 30°

The large experimental space, magnetoresistance at different temperatures, magnetic fields, and different tilt angles away from the b-axis present is not well discussed, and limits the readership.

We have extensively re-written the discussion and analysis of our data. We hope that it will ease the readership and convince the reviewer of the potential impact of our results.

REVIEWER COMMENTS

Reviewer #1 (Remarks to the Author):

The manuscript "Field-induced compensation of magnetic exchange as the origin of superconductivity above 40 T in UTe₂" is a revised manuscript resubmitted after it was reviewed and considered for publication in Nature Communications last year. I also reviewed the previous version of the manuscript. My comments on the previous version have been addressed and I recommend publication in Nature Communications with some revisions to the new content outlined below.

The paper presents experimental data (torque magnetization and electrical transport in high DC and pulsed fields) on UTe₂, a material of great current interest for its superconducting phases in combination with topologically nontrivial bandstructure. The authors present angle-dependent, temperature-dependent, and field-dependent data up to 70 T, allowing them to draw a phase diagram with focus on the high field superconducting phase in UTe₂. The experiments allow the determination of an extremely high critical field of 73 T, which is noteworthy, but the results presented in this paper have relevance beyond setting a record for the critical field in a superconductor. Experiments at such high fields are extremely challenging, and the high-quality data presented in this manuscript allow for a quantitative analysis of phase boundaries, coupling strength, and properties of the Fermi surfaces hosting the various superconducting phases in UTe₂.

The authors discuss a possible mechanism (Jaccarino-Peter effect) for the high-field reentrant superconducting phase and argue that it may be closely related to the low field superconducting phase (indicated by similar Fermi velocities), and that even conventional spin-singlet pairing may be possible in these superconducting phases that survive magnetic fields beyond the Pauli limit. This is in contrast with a popular hypothesis that links the field-reinforced phase at low tilt angles (believed to be a spin-singlet phase) to the high-field superconducting phase. This discussion was added to the paper since the first submission, and I find that it adds value, but the argumentation is at times difficult to follow. It may be helpful to restructure the discussion, starting with briefly outlining the different hypotheses for the various superconducting phases in UTe₂ and clearly stating how the conclusions in this manuscript differ from the existing discourse on UTe₂.

Beyond these general remarks, I have a few editorial comments / suggestions:

Page 5, near top:

"the volume of the sample is small enough in order to restrict the maximum torque... to safe value" -> "small enough to limit the max. torque to a safe value"

Page 6, end of first paragraph:

"However, we show its extend" -> "its extent" or "that it extends"

Page 7, near top:

"We mark the points used for the determination of the temperature dependence of the critical fields ..." -> "The critical fields at different temperatures were determined from the inflection points of the magnetoresistance curves"

Page 7, 3rd paragraph:

"Figure 2c shows the superconducting upper critical field..." -> clarify that this is the critical field for the low field superconducting phase

Page 10, near top:

"The overall slope of ρ_{xy} changes from positive to negative at highest fields" -> I am not convinced that the two traces in Fig. 3b show this slope change, as it is minimal compared to the noise.

Page 11, top:

"prevents to observe the effect" -> change to "prevent the observation" or "prevents us from observing". Same line: replace "cleary" with "clearly"

Page 12, near top:

"our understanding of mechanisms behind the AHE has experience a significant step forward..." -> "our understanding... has significantly improved"

Page 12, final sentence of top paragraph:

"the steep decline of the iAHE contribution suggests a suppression of the topological anomalies in the bands" -> perhaps discuss this in a little more detail or leave out?

Page 13, top : delete "(see below)"

Page 14, near top: "(see Fig 2e)" -> there is no panel labeled "e" in Fig 2.

Page 15, near top: "...is constant and parallel to the H" -> "is constant and parallel to the applied field H".

Fig 2: the colors in panel a are hard to distinguish (especially 1.4 K and 4.2 K), even though the overall temperature evolution is clear. Panel e is not labeled as panel e.

Fig 3b: it would be helpful to mark the fields corresponding to the metamagnetic and superconducting transitions, respectively.

Fig 4, caption (near end):

"in both cases as explained" -> "in both cases is explained"

Reviewer #2 (Remarks to the Author):

The authors have made significant improvements to the manuscript, addressing many of the issues with the writing. However, I remain unconvinced that the manuscript is suitable for publication in Nature Communications. The authors have attempted to link their findings to the mechanism of the high field superconducting state of UTe₂. If well supported, this will be very important and timely results. However, I feel that there is insufficient evidence to support this hypothesis, as stated below. Therefore, I believe that the manuscript has good results that merit publication but would be better suited for a more specialized journal, such as npj Quantum Materials or PRB.

The main point of the revised manuscript, as suggested by the title, is to propose the Jaccarino-Peter effect as the pairing mechanism for the high field superconducting state of UTe₂. This is partly in response to my previous critique that the manuscript lacked content that would be of interest to a broader audience. However, the authors' claim is not sufficiently supported by concrete evidence. They primarily base their assertion on the angle dependence of the normal state Hall resistance data, which exhibits a minimum at 30 degrees. Here is the authors' logic behind the argument: 1. Magnetization and resistance do not show suppression around 30 degree, 2. Analysis of the Hall data based on skew scattering mechanism does not show evidence for suppression of density of state, 3. Based on 1 and 2, author concludes that the Hall is dominated by intrinsic mechanism, namely change of topological anomaly, 4. One possibility along the line of intrinsic mechanism is the strong decrease of band polarization, 5. Such possibility can be consistent with the rest of their data, e.g., fit of H_{c2} of high field SC phase. This logic, in my opinion, lacks adequate justification. Obviously, as authors suggest,

strong decrease of band polarization is only one possibility. Again, as authors said in the paper, the Fermi surface in the Hm phase is still unknown, and therefore, the influence of the band structure topological of Hm on Hall is unknown. Without further evidence, it is too early to draw solid conclusions. Even the logic #3 is not well justified. The Hall effect is a complex phenomenon with different contributions. Author mentioned skew scattering and intrinsic Berry curvature effects, each of which has complicated dependencies on resistivity, magnetization, and other factors. In addition, as the angle changes, the data shifts from pure Hall effect to pure planer Hall effect, which could potentially lead to some minimum. The mechanism for planer Hall is even more complicated and is not discussed in the manuscript at all. In most manuscripts that discuss mechanisms of Hall effect, either scaling analysis or/and comparison to theoretical calculations needs to be performed. In the current case, without temperature dependence of Hall data, it is not easy to perform the scaling analysis. Also, the theoretical understanding for the Fermi surface of Hm phase has not been established, making the comparison with theory not realistic. Therefore, in my opinion, no solid conclusion on the mechanism of the Hall effect based on the current data can be made at all.

Again, I think there are undoubtedly good results in the manuscript. However, the hypothesis authors made based on the results is not well justified. At the moment, I can not recommend it for publications on Nature communications.

Reviewer #3 (Remarks to the Author):

The manuscript is extensively rewritten and adds the Jaccarino-Peter effect for discussing the re-entrant superconducting behavior.

The manuscript has several aims: first, it shows that FIB machining does not affect the superconducting properties of UTe₂, and second, it invokes the Jaccarino-Peter effect as a possible mechanism for re-entrant superconductivity. The sample has been measured using torque magnetometry, resistivity and Hall effect measurements at pulsed magnetic fields up to 70 T, varying the sample tilt angle in the (b,c) plane.

The manuscript is still hard to read as the phase space analyzed is large, and the behavior of the samples is complex. A short synopsis of the expected behavior of resistivity, magnetoresistivity, and Hall effect due to the Jaccarino-Peter effect and orbital limitation may be presented in the beginning to help in the description of the complex behavior.

With the jump at the MMT, a change in the spin alignment is observed. What are the expected demagnetization terms due to the sample shape at the different tilt angles along different orientations? Are there additional terms that decrease the effective magnetic field due to the sample shape?

The data presented in figure S1b shows an asymmetry in the torque for a temperature of 1.5K and angles around +/- 50 degrees. Is there a similar asymmetry at T=0.7K? Why is there an asymmetry?

Figure 2b represents the essence of the data measured. Is the hfsc area of existence dependent on the sample shape?

Page 36: "In UTe₂, the crystallographic orientation [011] seems specific in many respects: It is the natural cleavage plane of single crystals of UTe₂...", The authors confuse crystallographic directions (given in [uvw]) and plane normal directions, which, for an orthorhombic system, are not parallel. Do the authors get the crystallographic directions correct?

REVIEWER COMMENTS

Reviewer #1 (Remarks to the Author):

The manuscript "Field-induced compensation of magnetic exchange as the origin of superconductivity above 40 T in UTe₂" is a revised manuscript resubmitted after it was reviewed and considered for publication in Nature Communications last year. I also reviewed the previous version of the manuscript. My comments on the previous version have been addressed and I recommend publication in Nature Communications with some revisions to the new content outlined below.

The paper presents experimental data (torque magnetization and electrical transport in high DC and pulsed fields) on UTe₂, a material of great current interest for its superconducting phases in combination with topologically nontrivial bandstructure. The authors present angle-dependent, temperature-dependent, and field-dependent data up to 70 T, allowing them to draw a phase diagram with focus on the high field superconducting phase in UTe₂. The experiments allow the determination of an extremely high critical field of 73 T, which is noteworthy, but the results presented in this paper have relevance beyond setting a record for the critical field in a superconductor. Experiments at such high fields are extremely challenging, and the high-quality data presented in this manuscript allow for a quantitative analysis of phase boundaries, coupling strength, and properties of the Fermi surfaces hosting the various superconducting phases in UTe₂.

The authors discuss a possible mechanism (Jaccarino-Peter effect) for the high-field reentrant superconducting phase and argue that it may be closely related to the low field superconducting phase (indicated by similar Fermi velocities), and that even conventional spin-singlet pairing may be possible in these superconducting phases that survive magnetic fields beyond the Pauli limit. This is in contrast with a popular hypothesis that links the field-reinforced phase at low tilt angles (believed to be a spin-singlet phase) to the high-field superconducting phase. This discussion was added to the paper since the first submission, and I find that it adds value, but the argumentation is at times difficult to follow. It may be helpful to restructure the discussion, starting with briefly outlining the different hypotheses for the various superconducting phases in UTe₂ and clearly stating how the conclusions in this manuscript differ from the existing discourse on UTe₂.

We thank the reviewer for the encouraging words and assessment. To improve the readability we revised the introduction. In particular we have added a few more details about the main findings to the last paragraph of the introduction.

We have also added a new heading cutting in two the discussion of the analysis of the Hall effect. We now specify at the beginning where what this part will be about: see on page 12. We included the following lines:

"Analysis of the Hall effect and connection with the hfsc phase"

"In the following, we propose a scenario for the reentrant hfsc phase, supported by the analyses of our high-field torque and Hall-effect results. We will show that the origin of the Hall effect above H_m should be revisited, arising most likely from an intrinsic topological contribution. Hence, suppression of the Hall effect is best explained by that of the band polarization, leading naturally to a Jaccarino-Peter mechanism for the hfsc phase."

Finally, before the discussion of the JP compensation mechanism, where we removed some secondary details, we made a brief synopsis of the present identification of the different superconducting phases in UTe_2 , as suggested also by referee 3 (page 14):

"Jaccarino-Peter compensation effect in UTe_2 "

"Before we discuss this Jaccarino-Peter (JP) mechanism, let us summarize briefly the present situation for the various superconducting phases in UTe_2 at ambient pressure. Below H_m , there is a consensus both experimental and theoretical that the ground state is spin triplet, with a B_{3u} or A_u symmetry. Finer details such as the nodal structure are still under debate [39]. Recently, several experiments have revealed a clear phase transition to another superconducting phase for field along the b axis above ≈ 15 T at low temperatures [19, 56, 57]. Theoretical proposals anticipated a transition to a B_{2u} symmetry (d vector with no component along the b axis). In contrast, thermodynamic experiments have revealed drastic changes between the two phases, suggesting a change of the pairing mechanism in addition to a symmetry change [19]. For the case of a B_{2u} state [56, 57], it has also been shown that a spin-singlet state can account for the observed strong broadening of the superconducting anomaly as well as its angular dependence [19]. Regarding the hfsc phase, as opposed to the phases below H_m , there are no theoretical models yet, and the common wisdom is that it should be spin-triplet to survive such high fields. The initial proposed mechanisms were a Jaccarino-Peter or a Lebed mechanism [12], dismissed or abandoned for that of a "Landau level superconductivity" [32, 58]. This last hypothesis is rather surprising when contrasted with the existence of the hfsc phase even in very dirty systems [58]. In the following, we propose a Jaccarino Peter scenario mainly relying on a spin-singlet state. As shown already for the field-reinforced sc phase along the b axis, spin-singlet SC is able to survive in such high fields supported by strong-coupling effects and the field-induced reinforcement of the pairing strength [19]."

Beyond these general remarks, I have a few editorial comments / suggestions:

Page 5, near top:

"the volume of the sample is small enough in order to restrict the maximum torque... to safe value" -> "small enough to limit the max. torque to a safe value"

Corrected.

Page 6, end of first paragraph:

"However, we show its extend" -> "its extent" or "that it extends"

Corrected.

Page 7, near top:

"We mark the points used for the determination of the temperature dependence of the

critical fields ..." -> "The critical fields at different temperatures were determined from the inflection points of the magnetoresistance curves"

Corrected.

Page 7, 3rd paragraph:

"Figure 2c shows the superconducting upper critical field..." -> clarify that this is the critical field for the low field superconducting phase

Corrected. We added "of the Ifsc phase".

Page 10, near top:

"The overall slope of ρ_{xy} changes from positive to negative at highest fields" -> I am not convinced that the two traces in Fig. 3b show this slope change, as it is minimal compared to the noise.

The negative slope is most obvious in the high-resolution data presented in Fig. 3c. It furthermore is clearly visible in b for 28 ° and in d at higher temperatures for 35°. So, we believe that our description of the evolution of the slope is robust. We have added a reference to these two figures (3c and 3d) in the text.

Page 11, top:

"prevents to observe the effect" -> change to "prevent the observation" or "prevents us from observing". Same line: replace "cleary" with "clearly"

Corrected.

Page 12, near top:

"our understanding of mechanisms behind the AHE has experience a significant step forward..." -> "our understanding... has significantly improved"

Corrected.

Page 12, final sentence of top paragraph:

"the steep decline of the iAHE contribution suggests a suppression of the topological anomalies in the bands" -> perhaps discuss this in a little more detail or leave out?

We agree with the reviewer that this sentence was a bit misleading. We therefore revised this part into:

" ... the steep decline of the iAHE contribution suggests a suppressed influence of the topological aspect in the band structure on the AHE."

Page 13, top: delete "(see below)"

Corrected. We removed that part.

Page 14, near top: "(see Fig 2e)" -> there is no panel labeled "e" in Fig 2.

Corrected. We revised the labels in Fig 2.

Page 15, near top: "...is constant and parallel to the H" -> "is constant and parallel to the applied field H".

Corrected.

Fig 2: the colors in panel a are hard to distinguish (especially 1.4 K and 4.2 K), even though the overall temperature evolution is clear. Panel e is not labeled as panel e.

We revised colors in Fig. 2a in order to improve the visibility. We also corrected the labels of subfigures d and e.

Fig 3b: it would be helpful to mark the fields corresponding to the metamagnetic and superconducting transitions, respectively.

We revised Fig. 3 b and added a second panel that presents the data without additional offsets. This may help the reader to trace the transitions. It however is difficult to pin down the H_m position in this data set as it is quenched by the superconducting phase in the case of 28° and 38° data.

Fig 4, caption (near end):

"in both cases as explained" -> "in both cases is explained"

We revised the sentence into:

"The field dependence of λ above H_m is based on the compensation mechanism in both cases as explained in the main text and in Supplementary Note 5."

Reviewer #2 (Remarks to the Author):

The authors have made significant improvements to the manuscript, addressing many of the issues with the writing. However, I remain unconvinced that the manuscript is suitable for publication in Nature Communications. The authors have attempted to link their findings to the mechanism of the high field superconducting state of UTe₂. If well supported, this will be very important and timely results. However, I feel that there is insufficient evidence to support this hypothesis, as stated below. Therefore, I believe that the manuscript has good results that merit publication but would be better suited for a more specialized journal, such as npj Quantum Materials or PRB.

The main point of the revised manuscript, as suggested by the title, is to propose the Jaccarino-Peter effect as the pairing mechanism for the high field superconducting state of UTe₂. This is partly in response to my previous critique that the manuscript lacked content that would be of interest to a broader audience. However, the authors' claim is not sufficiently supported by concrete evidence. They primarily base their assertion on the angle dependence of the normal state Hall resistance data, which exhibits a minimum at 30 degrees. Here is the authors' logic behind the argument: 1. Magnetization and resistance do not show suppression around 30 degree, 2. Analysis of the Hall data based on skew scattering mechanism does not show evidence for suppression of density of state, 3. Based on 1 and 2, author concludes that the Hall is dominated by intrinsic mechanism, namely change of topological anomaly, 4. One possibility along the line of intrinsic mechanism is the strong decrease of band polarization, 5. Such possibility can be consistent with the rest of their data, e.g., fit of H_{c2} of high field SC phase. This logic, in my opinion, lacks adequate justification. Obviously, as authors suggest, strong decrease of band polarization is only one possibility. Again, as authors said in the paper, the Fermi surface in the H_m phase is still unknown, and therefore, the influence of the band structure topological of H_m on Hall is unknown. Without further evidence, it is too early to draw solid conclusions. Even the logic #3 is not well justified. The Hall effect is a complex phenomenon with different contributions. Author mentioned skew scattering and intrinsic Berry curvature effects, each of which has complicated dependencies on resistivity, magnetization, and other factors. In addition, as the angle changes, the data shifts from pure Hall effect to pure planer Hall effect, which could potentially lead to some minimum. The mechanism for planer Hall is even more complicated and is not discussed in the manuscript at all. In most manuscripts that discuss mechanisms of Hall effect, either scaling analysis or/and comparison to theoretical calculations needs to be performed. In the current case, without temperature dependence of Hall data, it is not easy to perform the scaling analysis. Also, the theoretical understanding for the Fermi surface of H_m phase has not been established, making the comparison with theory not realistic. Therefore, in my

opinion, no solid conclusion on the mechanism of the Hall effect based on the current data can be made at all.

In our work, we demonstrate a direct correlation between the angle dependence of H_{c2} and the Hall resistivity. The steep suppression is counter to what is expected for basic skew-scattering mechanisms. This mechanism has been suggested to be the dominant mechanism behind the unconventional temperature dependence of $4f$ and $5f$ heavy-fermion materials in the past. However, the analyses neglected potential contributions from intrinsic contributions that may not be identifiable by the temperature-dependence that easily. We understand the reviewer's suggestion to provide a more detailed T dependence analysis.

We however do not see how this could add more insights on the mechanism behind the unusual Hall response. Moreover, it is known that the comparison to predictions require not a T dependence, but a sample (residual resistivity) dependence (see e.g. the review of N. Nagaosa, ref. 42). We agree that a proper theoretical model for the electronic band structure in the high-field state would be extremely desirable for a profound understanding of the effect of H_m onto the Fermiology and the magnetic ground state behind that transition. It remains a challenging task to future experiments.

The idea of a contribution of the planar Hall effect is interesting, but most likely irrelevant for the geometry of our experiment. Indeed, the magnetization jump above H_m could play the same role as the ferromagnetic magnetization in systems showing large planar Hall effects. Nevertheless, as demonstrated by our torque measurements and discussed in the main text, this magnetization jump remains dominated by its b component, which for our sample geometry does not contribute to the planar Hall effect. Hence, we do not expect to have a significant contribution from this mechanism to ρ_{xy} in our experiment.

We recently, came across the work by Frank et al. [arXiv:2304.12392 (2023)]. In this work, three potential mechanisms for the high-field phase were listed. It is very speculative and the authors draw the wrong conclusions for the JP scenario from what has been reported for organic superconductors in the past. We mention the difference in our main text. The other two proposed scenarios are based on almost no grounds. There is no indication for quantum critical superconductivity and Landau-level stabilization is by far not realizable in the dirty system UTe_2 .

We therefore at this point rate our suggested mechanism as a first proposal for a mechanism for the hfsc phase, moreover supported by an independent physical property from the normal state (suppression of the Hall effect). Of course, we do not rule out alternative scenarios: the (short) history of UTe_2 has already been full of a rich variety of scenarios for its superconducting states, soon dismissed by the next generation of experiments or crystals. However, we do believe that our work contributes to a tough nut to crack and provides unprecedented insights on its behavior within an experimentally and theoretically extremely challenging parameter space. The overall extremes and the theoretical implications make this topic interesting to a very broad readership.

Again, I think there are undoubtedly good results in the manuscript. However, the hypothesis authors made based on the results is not well justified. At the moment, I can not recommend it for publications on Nature communications.

We thank the referee for her/his positive appreciation of our results, and we hope to have convinced her/him that our hypothesis is significantly more solid than all proposals put forward up to now for the hfsc phase, even though we are well aware that it is not a definite proof: it is, however, solid enough to stimulate further work challenging this proposal.

Reviewer #3 (Remarks to the Author):

The manuscript is extensively rewritten and adds the Jaccarino-Peter effect for discussing the re-entrant superconducting behavior.

The manuscript has several aims: first, it shows that FIB machining does not affect the superconducting properties of UTe₂, and second, it invokes the Jaccarino-Peter effect as a possible mechanism for re-entrant superconductivity. The sample has been measured using torque magnetometry, resistivity and Hall effect measurements at pulsed magnetic fields up to 70 T, varying the sample tilt angle in the (b,c) plane.

The manuscript is still hard to read as the phase space analyzed is large, and the behavior of the samples is complex. A short synopsis of the expected behavior of resistivity, magnetoresistivity, and Hall effect due to the Jaccarino-Peter effect and orbital limitation may be presented in the beginning to help in the description of the complex behavior.

As suggested also by the first referee, we tried to improve the readability of the paper (see answer to referee 1 for more details):

In order to introduce the topic to a broader readership a bit better, we revised the beginning of the introduction into:

"Superconductivity is notoriously fragile under magnetic field, all the more when the superconducting critical temperature is small. However, the sensitivity of superconductors to magnetic fields varies greatly conditioned by various influences. For example, a whole class of strongly correlated electron systems called "heavy fermions" exhibits critical fields several orders of magnitude larger than other superconducting systems with similar T_c (usually sub-Kelvin), precisely because the quasi particles possess heavy effective masses, or equivalently, very slow Fermi velocities [1–3]. In many heavy fermions, the upper critical field is limited at low temperatures by the paramagnetic limit that arises from the Zeeman coupling of the Cooper pair spins to the external field [4, 5]. In other superconductors, only a strong 2D character may allow for enhanced upper critical fields close to that limit. The recent discovery of superconductivity (SC) in the heavy-fermion metal UTe₂ [6] with a critical temperature

T_c ≈ 2 K, triggered much excitement, as its critical field reaches values approaching those of high-T_c superconductors."

We added already in the introduction a few more details into the last paragraph of the paper in order to guide the reader a bit better to the main content of the paper. We also made sure to stay coherent with the labelling of the various phases that are discussed in this work.

We have divided the long analysis section of the Hall effect in to two parts, announcing at the beginning where we were heading to. We also revised the JP compensation section with the focus to reduce its complexity and improve the readability, and added a sentence to make it clearer why the behavior of the Hall effect might reflect the JP compensation mechanism (top of page 15):

"If H and H_{ex} compensate each other, then the itinerant quasiparticles feel no Zeeman field and they should lose their polarization: Our Hall-effect results suggest that in the angular range around 30°..."

We refer to the supplement for further details about the mechanism and modelling.

With the jump at the MMT, a change in the spin alignment is observed. What are the expected de-magnetization terms due to the sample shape at the different tilt angles along different orientations? Are there additional terms that decrease the effective magnetic field due to the sample shape?

The question of demagnetization field is of course legitimate. However, we expect a very tiny correction, below 0.2% on the applied field in UTe₂. Indeed, the distance between uranium atoms is rather large in UTe₂, and with a moment of 1μ_B per uranium, μ₀M is of the order of 0.13 T. Hence, for a field perpendicular to an "infinite cylinder" as an approximation of our sample shape, the correction should be ½μ₀M = 0.07 T at 40 T. This is beyond the accuracy of the field measurement! Variations of the factor between the two perpendicular directions (current along a, field in the b,c plane) would be even smaller.

The data presented in figure S1b shows an asymmetry in the torque for a temperature of 1.5K and angles around +/- 50 degrees. Is there a similar asymmetry at T=0.7K ? Why is there an asymmetry?

The asymmetry is an experimental artefact. It originates from the asymmetric stiffness of the piezoelectric microcantilever, built in by the piezoresistive thin film deposited onto one side. It becomes most apparent for larger deflections. Hence, this asymmetry is also expected for other temperatures. We have added a Note into the Figure caption, saying:

"The asymmetry between positive- and negative-angle curves is caused by the asymmetric deflection stiffness of the piezoresistive microcantilever."

Figure 2b represents the essence of the data measured. Is the hfsc area of existence dependent on the sample shape?

The FIB shaped devices #1 and #2 had different cross sections. We do not observe significant differences in the overall superconducting properties for these devices, and the absence of an effect connected to the sample shape is naturally consistent with the weakness of the demagnetization corrections.

Page 36: "In UTe_2 , the crystallographic orientation $[011]$ seems specific in many respects: It is the natural cleavage plane of single crystals of UTe_2 ...", The authors confuse crystallographic directions (given in $[uvw]$) and plane normal directions, which, for an orthorhombic system, are not parallel. Do the authors get the crystallographic directions correct?

All angles given in our work were determined in the framework span by the (b,c) lattice directions. We, therefore, are confident about the stated orientations. That means the (001) and (010) directions. We revised the mentioned sentence into:

"In UTe_2 , the crystallographic orientation $[011]$ seems specific in many respects: The (011) plane is a natural cleavage plane of single crystals of UTe_2 ,"

More precisely in real space, the normal to the (011) "cleavage" plane is approximately at 23.7° from the b axis in the b,c plane.

Reviewers' comments:

Reviewer #1 (Remarks to the Author):

The paper "Field-induced compensation of magnetic exchange as the origin of superconductivity above 40 T in UTe₂" by T. Helm and coworkers reports on magnetic and electrical transport measurements of superconducting phases in UTe₂, most notably the high-field superconducting phase with a record-breaking upper critical field exceeding 70 T.

I reviewed two previous versions of the manuscript. In this version, the authors extended the discussion of a possible mechanism (Jaccarino-Peter Effect) that could explain the extremely high critical fields and is consistent with other experimental observations. They also added context about the current theoretical understanding of the various superconducting phases in UTe₂, which is generally helpful and highlights the relevance of the results presented in this paper.

I recommend publication in Nature Communications with minor editorial revisions. Below are some suggestions for minor changes to the first paragraph:

"Superconductivity is notoriously fragile under magnetic field" -> magnetic fields

"However, the sensitivity of superconductors to magnetic fields greatly conditioned by various influences" -> sensitivity [...] to magnetic fields is influenced by a variety of factors.

"In many heavy fermions" -> in many heavy fermion materials

Reviewer #2 (Remarks to the Author):

In the revised manuscript the issues I raised remain to be there, with not much has been changed. Again, I do not feel that authors have concrete evidence to link the suppression of Hall resistivity to the Jaccarino-Peter mechanism of the high field induced superconductivity in UTe₂. This hypothesis heavily relies on their interpretation of the origin of the Hall effect, which they attributed to Berry curvature effect. In the rebuttal letter, the authors claim, "Moreover, it is known that the comparison to predictions require not a T dependence, but a sample (residual resistivity) dependence (see e.g. the review of N. Nagaosa, ref. 42)." I am afraid that this is not completely right. The basic idea to distinguish the skew scattering contribution and Berry curvature contribution is see the dependence of the Hall resistivity on the scattering rate. This can be done by comparing samples with different residual resistivity, but also can be done via the scaling analysis which requires temperature dependence. The sample dependence is mainly seen in paper a decade ago when the scaling analysis was not fully established. In more recent papers, scaling analysis is more popular. Here are some of references where people conclusively established the mechanism of the Hall resistivity: theory papers on the scaling analysis: Rev. Lett. 103, 087206 (2009), Phys. Rev. Lett. 114, 217203 (2015); famous experimental example of Berry curvature contribution: Nature 555, 638–642 (2018), Nat. Phys. 14, 1125–1131 (2018); experimental example that shows skew scattering mechanism: ACS Nano 15, 9759–9763 (2021). A similar analysis is required if authors want to make a serious claim of the origin of the Hall resistivity. I am also totally fine if they can establish this firmly with sample dependence since they seem to prefer this old fashion. But I don't find that either in their manuscript.

Upon closer examination, I also found another problem with their claim, which provides another alternative to explain their Hall data. In the manuscript, the authors claim that "Similarly, the resistivity does not show significant changes around H_m for both field orientations [17], indicating that neither the elastic nor the inelastic scattering display a considerable evolution with angle." However, this is not so true according to their data. Fig 1d clearly shows that at 0.7 K (which is the same temperature for the Hall resistivity plot on Fig 4a) resistivity shows a clear minimum at 35 degrees, the exact same trend as their Hall resistivity from the same sample #1. Authors do not have a similar

plot for sample #2. But based on their SI Fig S3, one can estimate. My rough estimation shows that for sample #2 resistivity, at 0.7 K and 65 T, shows very similar trend as their Hall resistivity, basically a minimum at 35 degrees. This strongly indicates that the angle dependence of their Hall resistivity is dominated by the anisotropy of the magnetoresistance. There is no need to refer to the band effect. To summarize, I do not find their claim of the intrinsic mechanism of the Hall resistivity to be firmly supported. Therefore their claim about the JP mechanism of high field induced superconductivity does not have solid support either. I can not recommend the manuscript for publication on Nature Comm.

Reviewer #3 (Remarks to the Author):

The manuscript is much improved and reads better.

A minor point: the torque measurements are less quantitative due to the fact that the superconducting screening currents are dependent on the shape of the sample.

The strong anisotropy of H_{c2} in the hfsc state should be quantified and entered into a table, for instance in the supplement.

To what extent would multi band superconductivity account for the strong anisotropy in H_{c2} ? Was multi band superconductivity considered?

Point-by-point response:

Reviewers' comments:

Reviewer #1 (Remarks to the Author):

The paper "Field-induced compensation of magnetic exchange as the origin of superconductivity above 40 T in UTe₂" by T. Helm and coworkers reports on magnetic and electrical transport measurements of superconducting phases in UTe₂, most notably the high-field superconducting phase with a record-breaking upper critical field exceeding 70 T.

I reviewed two previous versions of the manuscript. In this version, the authors extended the discussion of a possible mechanism (Jaccarino-Peter Effect) that could explain the extremely high critical fields and is consistent with other experimental observations. They also added context about the current theoretical understanding of the various superconducting phases in UTe₂, which is generally helpful and highlights the relevance of the results presented in this paper.

I recommend publication in Nature Communications with minor editorial revisions.

>> We thank the referee for the positive assessment and the recommendation for publishing our work in Nature Communication. <<

Below are some suggestions for minor changes to the first paragraph:

"Superconductivity is notoriously fragile under magnetic field" -> magnetic fields

>> We prefer to stick with the singular version. <<

"However, the sensitivity of superconductors to magnetic fields greatly conditioned by various influences" -> sensitivity [...] to magnetic fields is influenced by a variety of factors.

"In many heavy fermions" -> in many heavy fermion materials

>> We have taken these editorial suggestions into account and modified the text accordingly. <<

Reviewer #2 (Remarks to the Author):

In the revised manuscript the issues I raised remain to be there, with not much has been changed. Again, I do not feel that authors have concrete evidence to link the suppression of Hall resistivity to the Jaccarino-Peter mechanism of the high field induced superconductivity in UTe₂. This hypothesis heavily relies on their interpretation of the origin of the Hall effect, which they attributed to Berry curvature effect. In the rebuttal letter, the authors claim, “Moreover, it is known that the comparison to predictions require not a T dependence, but a sample (residual resistivity) dependence (see e.g. the review of N. Nagaosa, ref. 42).” I am afraid that this is not completely right. The basic idea to distinguish the skew scattering contribution and Berry curvature contribution is see the dependence of the Hall resistivity on the scattering rate. This can be done by comparing samples with different residual resistivity, but also can be done via the scaling analysis which requires temperature dependence. The sample dependence is mainly seen in paper a decade ago when the scaling analysis was not fully established. In more recent papers, scaling analysis is more popular. Here are some of references where people conclusively established the mechanism of the Hall resistivity: theory papers on the scaling analysis: Rev. Lett. 103, 087206 (2009), Phys. Rev. Lett. 114, 217203 (2015); famous experimental example of Berry curvature contribution: Nature 555, 638–642 (2018), Nat. Phys. 14, 1125–1131 (2018); experimental example that shows skew scattering mechanism: ACS Nano 15, 9759–9763 (2021). A similar analysis is required if authors want to make a serious claim of the origin of the Hall resistivity. I am also totally fine if they can establish this firmly with sample dependence since they seem to prefer this old fashion. But I don’t find that either in their manuscript.

<< Again, the referee seems to miss the major complication of UTe₂ when it comes to the determination of the Hall components.

The experimental references, suggested by the referee, do not have to deal with the extreme parameters of field, temperature and orientation. Furthermore, the additional (still unknown) metamagnetic transition in high fields at H_M complicates a potential scaling analysis. We know the standard way of how to disentangle the contributions in the Hall effect very well. We, however, need to note again, that a detailed scaling analysis of the Hall response has been provided previously in Reference Niu et al. [PRR 2, 033179 (2020)]. The authors (including some of the coauthors of the present work) showed that the Hall signal in UTe₂ is comprised of a significant anomalous contribution. The standard analysis of the high-field hall coefficient revealed a sudden jump at H_M, see figure below:

However, the distinction between an intrinsic mechanism and skew scattering has been a challenge. It is complicated by the non-saturating magnetization in the high-field state of UTe₂. Therefore, a direct determination of the orbital coefficient from the field-saturated Hall slope is not possible. In the previous work, we concluded that the major contribution is associated with skew scattering due to 5f-electrons following the consensus for various 4f and 5f compounds. Here, the scaling of the Hall resistivity with the square of the longitudinal resistivity was taken as evidence for this contribution, see for example Onuki et al. [JPSJ 58, 2119 (1989)] in the figure below :

[redacted]

However, at the time these conclusions were made in the past, the overall theory of Hall was less evolved, as it was lacking the Berry curvature formalism.

Only in recent years, the intrinsic component could be quantitatively determined for some materials. We, therefore, suggest that the previous conclusion by Niu et al. may be revisited. The jump in the orbital Hall coefficient around H_M by a factor of 10 has not been confirmed by other quantities. For example, Miyake et al. conducted magnetization measurements at various field orientations and temperatures. They concluded that the γ value jumps only slightly at H_M signaling a much weaker change in the electronic structure than expected from the Hall analyses. Hence, this provides another hint for the potential intrinsic Hall that is induced by the new magnetic ground state

above H_M . We furthermore would like to mention again that the magnetization sees no saturation above H_M and our magnetic torque experiments indicate that a significant perpendicular component persists beyond the metamagnetic transition. Band-structure calculations may be able to estimate the intrinsic contribution to the Hall. Therefore, it would be highly desirable to obtain spectroscopic insights for the field-induced state above H_M by, e.g., ARPES or magnetic-quantum-oscillation studies. >>

Upon closer examination, I also found another problem with their claim, which provides another alternative to explain their Hall data. In the manuscript, the authors claim that “Similarly, the resistivity does not show significant changes around H_m for both field orientations [17], indicating that neither the elastic nor the inelastic scattering display a considerable evolution with angle.” However, this is not so true according to their data. Fig 1d clearly shows that at 0.7 K (which is the same temperature for the Hall resistivity plot on Fig 4a) resistivity shows a clear minimum at 35 degrees, the exact same trend as their Hall resistivity from the same sample #1. Authors do not have a similar plot for sample #2. But based on their SI Fig S3, one can estimate. My rough estimation shows that for sample #2 resistivity, at 0.7 K and 65 T, shows very similar trend as their Hall resistivity, basically a minimum at 35 degrees. This strongly indicates that the angle dependence of their Hall resistivity is dominated by the anisotropy of the magnetoresistance. There is no need to refer to the band effect.

<< Thank you for the careful consideration of the additional data we are providing. Indeed, there seems to be a shallow minimum in the longitudinal magnetoresistivity that coincides with the minimum in the Hall effect. This we also mention in the main text. However, the overall reduction of MR with angle is small and cannot account for a complete vanishing of the Hall signal as observed. As was shown previously, the anomalous component in the Hall scales with the square of the longitudinal resistivity and the magnetization. The latter does not exhibit significant changes upon rotation and, in particular, the jump at H_M remains steady through the 30 ° region. The mentioned anisotropy in the magnetoresistance, therefore, cannot account for the vanishing of the overall Hall at around 35 °. >>

To summarize, I do not find their claim of the intrinsic mechanism of the Hall resistivity to be firmly supported. Therefore their claim about the JP mechanism of high field induced superconductivity does not have solid support either. I can not recommend the manuscript for publication on Nature Comm.

>>We strongly disagree with this conclusion. Our work provides a substantial advancement on the subject and presents multiple firsts and new results. We report the very first successful magnetic torque experiments in pulsed magnetic fields on UTe_2 single crystals. We report the very first transport results with microfabricated devices that provide the highest resolution achievable to date. We provide the first in-depth investigation of the angular dependence of H_{c2} in the high-field state and the high-field Hall resistivity. Even though a quantitative estimate of an intrinsic Hall component cannot be provided at this point, we provide clear evidence for the vanishing of the

anomalous component in UTe₂ and demonstrate a clear correlation with the H_{c2} angle dependence. We are able to describe our results by a model based on the Jaccarino-Peter mechanism that appears as the most likely mechanism for the emergence of high-field superconductivity.

The suggested scaling analysis has been performed for H||b previously and revealed a significant jump in the Hall coefficient associated to the orbital Hall component. A clear scaling of the Hall signal with the square of the longitudinal resistivity was observed. In cerium and uranium based heavy-fermion compounds this has been associated with a dominant skew-scattering mechanism [see references 42-44 in the main text]. However, without spectroscopic knowledge of the magnetic structure in the field-induced state it is challenging to disentangle a potential intrinsic from skew-scattering component. This point we explain in the paper. We, furthermore, stress that UTe₂ should be dominated by the intrinsic effect as it exhibits moderate-metallic properties and hence, would not match the empirically determined conductivity values expected for the skew-scattering dominated Hall regime. [see Nagaosa, Rev. Mod. Phys. 82 (2010)]. >>

Reviewer #3 (Remarks to the Author):

The manuscript is much improved and reads better.

>> We thank the referee for the previous suggestions and the positive assessment after the latest changes. <<

A minor point: the torque measurements are less quantitative due to the fact that the superconducting screening currents are dependent on the shape of the sample.

>> We agree with the referee. As our experiments were performed in pulsed magnetic field, we do not expect significant signatures of superconductivity in the magnetic torque. If at all, the effect would be minor. The overall torque amplitude would not be useable for quantitative estimations of that contribution. As we discussed in the previous response we provide a rough estimation of the demagnetization effect and rate it a minor effect.

>>

The strong anisotropy of H_{c2} in the hfsc state should be quantified and entered into a table, for instance in the supplement.

<< Thank you for this suggestion. We included a new Figure (Fig. S4) into our supplementary information.

It compares the angle-dependent anisotropy of H_{c2} in the hfsc with the lfsc phase. We also mention this at the end of section “ H_{c2} in the field-induced reentrant hfsc phase” in the main text. >>

To what extent would multi band superconductivity account for the strong anisotropy in H_{c2} ? Was multi band superconductivity considered?

<< As recently observed by magnetic quantum oscillations and confirmed by band theory the zero-field band structure of UTe₂ consists of at least two different electron- and hole-type 2D Fermi sheets. Hence, there is potential room for multiband superconductivity in this compound. >>

REVIEWER COMMENTS

Reviewer #1 (Remarks to the Author):

Thank you for the opportunity to review the manuscript "Evidence for field-induced compensation of magnetic exchange as the origin of superconductivity above 40 T in UTe₂" by T. Helm and coworkers. I had already recommended the manuscript for publication and was specifically asked to review Reviewer 2's comments and the authors' rebuttal.

Reviewer 2's primary criticism is that in their opinion the data doesn't provide sufficient evidence for field-induced compensation (Jaccarino-Peter effect) as the mechanism behind the high field superconducting phase in UTe₂. I agree that this interpretation relies heavily on the anomalous Hall effect (AHE) data, and as stated by both the reviewer and the authors, distinguishing between the skew scattering and Berry curvature AHE components isn't possible based on the measurements presented in the paper. However, the data does show a convincing suppression of the Hall resistivity (Fig 4), where I agree with the authors that it can't be explained simply from anisotropy of the magnetoresistance (as suggested by Reviewer 2). In my opinion, the data is consistent with a Jaccarino-Peter mechanism, however definitive proof would require more data on the magnetic ground state, as also stated by the authors.

In my opinion, the greatest value of this work lies not in proving a field-induced compensation as the origin of superconductivity, but presenting a systematic, coherent, high-quality dataset on both magnetic torque and magnetotransport of UTe₂ under extremely high fields at low temperatures, and establishing the phase boundary. Showing that the Hall effect goes to zero is in itself a highly valuable piece of information to demonstrate a superconducting phase. This is primarily an experimental paper, and the data presented will form the basis of future discussions and additional experiments and theoretical work to fully understand the high field superconducting phase, but it needs to be published first. To be clear, I think the authors go far beyond presenting a purely phenomenological collection of data, and the discussion about the field-induced compensation mechanism definitely added value to the paper, but it is not the main reason I find it worthy of publication in Nature Communications. If a wealth of experimental data on UTe₂ already existed in this field and temperature range and this paper was aimed at establishing the mechanism behind the high field superconducting phase beyond a reasonable doubt, I would side with Reviewer 2 and suggest that additional data, e. g. on the magnetic ground state, are necessary to make this argument. However, there isn't nearly enough published high-field data on UTe₂ to settle this discussion at this point, and this paper will not settle it, but more importantly it provides a wealth of information to even start it.

I definitely recommend publication in Nature Communications, but may suggest changing the title to "Field-induced compensation of magnetic exchange as the possible origin of superconductivity above 40 T in UTe₂" to be a little more diplomatic.

Reviewer #4 (Remarks to the Author):

Referee Report UTe₂

Referee Report:

Having read the revised manuscript, the reports of the 3 referees and the rebuttal of the authors, I conclude that the manuscript is a nice piece of detective work and makes a compelling case for the validity of Jaccarino-Peter effect with regards to the field-induced phases of UTe₂. Moreover, it is an exceptional study involving micron-sized samples, which means they are better able to eliminate sample heating as a dominant effect than in earlier studies made at high magnetic fields. This also

enabled careful torque and transport measurements. As an additional bonus, the authors find that the higher field-induced superconducting phase has an upper critical field close to 69T, or 73T upon extrapolation.

A crucial part of the study is that magnetic torque measurements find the component of the magnetization orthogonal to the field to reach a minimum near 25 degrees. This sets up the necessary preconditions for the Jaccarino-Peter effect. The authors have additional evidence from Hall effect measurements. It is important to note that because the Fermi surface consists of small pockets, the large moment per site at high magnetic fields cannot simply result from the polarization of the pockets. In this way, the moments are external to the small pockets of Fermi surface and therefore play an analogous role to that of polarized magnetic moments in the Jaccarino-Peter effect. The Hall effect provides evidence for the compensation mechanism effecting the effective degree of polarization of the Fermi surface. It would be helpful if the authors made these points more clearly in their manuscript.

Referee 2 provides arguments for the authors being novices with regards to their analysis of the anomalous Hall effect, and for them not having not taken this properly into consideration. However, upon reading the manuscript and the rebuttal of the authors it becomes clear that some of the authors are seasoned experts in the anomalous Hall effect and describe a situation that is more complex than found in text book heavy fermion systems. The authors have explained themselves quite convincingly. However, if the manuscript is to be broadly received, the authors really need to take their reasoning from their rebuttal and make it available for the general reader as supplementary information text.

There are a number of other minor issues that should be addressed before the paper is accepted for publication:

- 1) Why not put "Evidence for the Jaccarino-Peter effect....." or something along those lines in the title?
- 2) In the second to last line of the abstract, shouldn't the word "possible" be replaced by "likely."
- 3) On page 4, "tailored" is a strange word to use. Why not use the word "cut" instead?
- 4) The authors should probably replace the word "monotonous" with "monotonic" throughout the manuscript, unless the authors really did fall asleep while doing the measurements.
- 5) On page 5, the description of the minimum in the magnitude of the jump in magnetic torque at 25 degrees is not at all well explained. It took me some time to examine the data in Fig. S1 to figure out what exactly the authors were referring to. Perhaps this is why referee 2 didn't find the paper very convincing? I recommend reworking the text here to make sure that this point is made very clearly. I would indicate the minimum with an arrow in the figure. I would also suggest putting Fig. S1 in the main text, since it's a key piece of thermodynamic evidence for the Jaccarino-Peter effect.
- 6) The last sentence in the "Magnetic torque transition" section needs to be made more clear. The authors should just plainly state that the component of the magnetization perpendicular to the field can be detected in magnetic torque measurements. This will avoid having to refer to prior magnetization measurements (which were perfectly reasonable) in a negative light.
- 7) The authors sometimes use "lfsc" and "hfsc" and other times use "lfSC" and "hfSC." The authors need to be consistent with their usage of lower and upper case letters; not alternate between the two.
- 8) Page 9 line 5. The authors should state that they are assuming that the same λ that is renormalizing the mass also determines the strength of the pairing. One has to be careful here, because there are multiple contributions (e.g. electron-electron and electron-phonon) to the mass enhancement.

9) In UPt₃ and UPd₂Al₃, it was found by way of de Haas-van Alphen measurements that the masses are just as heavy above H_m as they are below H_m. This is in contrast to what one typically finds in Ce compounds. When we consider the effective masses in UPt₃ and UPd₂Al₃, UTe₂ is no longer so unique with regards to the minimal changes in the Fermi surface at H_m.

10) Use of the word "triggering" at the top of page 10 is strange. I would recommend replacing this with "the subject of."

11) At the top of page 12, the Yamaji effect normally suppresses the interlayer conductivity in a layered system. Perhaps the authors meant "suppressed conduction" instead of "enhanced conduction?" Or maybe they are referring to the conductivity between Yamaji angles?

12) In regards to the bottom of page 13, there are papers out there still debating whether UTe₃ is spin triplet or not. Hence, they should not state that there is a consensus. Maybe they should state instead that there is consensus for UTe₃ being recognized as candidate spin triplet superconductor?

Point-by-point response to the reviewer comments.

REVIEWER

COMMENTS

Reviewer #1 (Remarks to the Author):

Thank you for the opportunity to review the manuscript "Evidence for field-induced compensation of magnetic exchange as the origin of superconductivity above 40 T in UTe₂" by T. Helm and coworkers. I had already recommended the manuscript for publication and was specifically asked to review Reviewer 2's comments and the authors' rebuttal.

Reviewer 2's primary criticism is that in their opinion the data doesn't provide sufficient evidence for field-induced compensation (Jaccarino-Peter effect) as the mechanism behind the high field superconducting phase in UTe₂. I agree that this interpretation relies heavily on the anomalous Hall effect (AHE) data, and as stated by both the reviewer and the authors, distinguishing between the skew scattering and Berry curvature AHE components isn't possible based on the measurements presented in the paper. However, the data does show a convincing suppression of the Hall resistivity (Fig 4), where I agree with the authors that it can't be explained simply from anisotropy of the magnetoresistance (as suggested by Reviewer 2). In my opinion, the data is consistent with a Jaccarino-Peter mechanism, however definitive proof would require more data on the magnetic ground state, as also stated by the authors.

In my opinion, the greatest value of this work lies not in proving a field-induced compensation as the origin of superconductivity, but presenting a systematic, coherent, high-quality dataset on both magnetic torque and magnetotransport of UTe₂ under extremely high fields at low temperatures, and establishing the phase boundary. Showing that the Hall effect goes to zero is in itself a highly valuable piece of information to demonstrate a superconducting phase. This is primarily an experimental paper, and the data presented will form the basis of future discussions and additional experiments and theoretical work to fully understand the high field superconducting phase, but it needs to be published first. To be clear, I think the authors go far beyond presenting a purely phenomenological collection of data, and the discussion about the field-induced compensation mechanism definitely added value to the paper, but it is not the main reason I find it worthy of publication in Nature Communications. If a wealth of experimental data on UTe₂ already existed in this field and temperature range and this paper was aimed at establishing the mechanism behind the high field superconducting phase beyond a reasonable doubt, I would side with Reviewer 2 and suggest that additional data, e. g. on the magnetic ground state, are necessary to make this argument. However, there isn't nearly enough published high-field data on UTe₂ to settle this discussion at this point, and this paper will not settle it, but more importantly it provides a wealth of information to even start it.

I definitely recommend publication in Nature Communications, but may suggest changing the title to "Field-induced compensation of magnetic exchange as the possible origin of superconductivity above 40 T in UTe₂" to be a little more diplomatic.

>>

We are grateful for this very positive review. We followed the reviewer's suggestion and added "possible" to the title.

<<

Reviewer #4 (Remarks to the Author):

Referee Report UTe₂

Referee Report:

Having read the revised manuscript, the reports of the 3 referees and the rebuttal of the authors, I conclude that the manuscript is a nice piece of detective work and makes a compelling case for the validity of Jaccarino-Peter effect with regards to the field-induced phases of UTe₂. Moreover, it is an exceptional study involving micron-sized samples, which means they are better able to eliminate sample heating as a dominant effect than in earlier studies made at high magnetic fields. This also enabled careful torque and transport measurements. As an additional bonus, the authors find that the higher field-induced superconducting phase has an upper critical field close to 69T, or 73T upon extrapolation.

>>

We are delighted to read that reviewer #4 shares our excitement and overall view on our results.

<<

A crucial part of the study is that magnetic torque measurements find the component of the magnetization orthogonal to the field to reach a minimum near 25 degrees. This sets up the necessary preconditions for the Jaccarino-Peter effect. The authors have additional evidence from Hall effect measurements. It is important to note that because the Fermi surface consists of small pockets, the large moment per site at high magnetic fields cannot simply result from the polarization of the pockets. In this way, the moments are external to the small pockets of Fermi surface and therefore play an analogous role to that of polarized magnetic moments in the Jaccarino-Peter effect. The Hall effect provides evidence for the compensation mechanism effecting the effective degree of polarization of the Fermi surface. It would be helpful if the authors made these points more clearly in their manuscript.

>>

Yes indeed, such an explanation was missing, and we thank the referee for this helpful remark. We modified accordingly the last paragraphs on page 13, concluding discussing the connection between Hall effect and hfSC phase, before the discussion on the Jaccarino Peter effect. It now reads:

“Band splitting with avoided level crossing is key for this intrinsic contribution to the Hall

effect [45]. So an appealing possibility is that the suppression of the iAHE contribution arises from a strong decrease of the band polarization in this angular range. It could result from a compensation between the applied field and an exchange field between the conduction bands and local magnetic moments, polarized by the metamagnetic transition. The background picture for this scenario is that a main contribution to the magnetization of UTe_2 arises from localized 5f-electrons. This is consistent with the large nearest-neighbor distance, far exceeding the Hill limit [56]. It is furthermore supported by band-structure calculations that predict a Fermi surface dominated by Te-5p and U-6d electrons (partly hybridized with U-5f), with at most only small 5f - electron pockets [26]. These, however, have not been observed by experiments to date [52]. In such a scheme, the jump of the magnetization at H_m arises mainly from local moments having (antiferromagnetic) exchange coupling with the conduction bands, a very natural scheme for a Kondo system. A reduction of the band polarization, arising from the compensation between exchange and applied field above H_m also explains why we can fit H_{c2} in the hfSC phase with the assumption of unaltered $\langle v^{\text{band}_F} \rangle$ values as compared to the lfSC phase (Fig. 2e.): the main effects of the metamagnetic transition on the Fermi surface then disappear. More importantly, this compensation between H and the “molecular” exchange field is instrumental for the so-called Jaccarino-Peter mechanism [57–59] that could account for the reentrant hfSC phase.”

<<

Referee 2 provides arguments for the authors being novices with regards to their analysis of the anomalous Hall effect, and for them not having not taken this properly into consideration. However, upon reading the manuscript and the rebuttal of the authors it becomes clear that some of the authors are seasoned experts in the anomalous Hall effect and describe a situation that is more complex than found in text book heavy fermion systems. The authors have explained themselves quite convincingly. However, if the manuscript is to be broadly received, the authors really need to take their reasoning from their rebuttal and make it available for the general reader as supplementary information text.

>>

We followed the recommendation of the reviewer and added the following explanation to the end of Supplementary Note 4:

“The general analyses of the Hall effect is complicated by the appearance of metamagnetism in UTe_2 , the lack of detailed information about the magnetic structure and its effects on the band structure of UTe_2 . In order to discriminate the orbital from

the anomalous coefficient an approach would be to study the high-field Hall effect in the field range where the magnetism is saturated so that the AHE contribution remains constant. As the magnetization of UTe_2 does not saturate above H_m up to 70 T such an approach could not be applied in our case”

<<

There are a number of other minor issues that should be addressed before the paper is accepted for publication:

1) Why not put “Evidence for the Jaccarino-Peter effect.....” or something along those lines in the title?

>>

In order to follow the suggestions of both Reviewer #1 and #4 we added the word “possible” to the title.

<<

2) In the second to last line of the abstract, shouldn't the word “possible” be replaced by “likely.”

>>

We followed this suggestion and modified the abstract accordingly.

<<

3) On page 4, “tailored” is a strange work to use. Why not use the word “cut” instead?

>>

We followed this suggestion and changed “tailored” to “cut” accordingly.

<<

4) The authors should probably replace the word “monotonous” with “monotonic” throughout the manuscript, unless the authors really did fall asleep while doing the measurements.

>>

Thank you for this correction. We have replaced Monotonous by monotonic throughout the text.

<<

5) On page 5, the description of the minimum in the magnitude of the jump in magnetic torque at 25 degrees is not at all well explained. It took me some time to examine the data in Fig. S1 to figure out what exactly the authors were referring to. Perhaps this is why referee 2 didn't find the paper very convincing? I recommend reworking the text here to make sure that this point is made very clearly. I would indicate the minimum with an arrow in the figure. I would also suggest putting Fig. S1 in the main text, since its a key piece of thermodynamic evidence for the Jaccarino-Peter effect.

>>

We modified Figure 1 (d) and included a magnetic-torque trace at fixed field of 60 T. We also updated the Figure caption and the part referring to this data on page 5. It now reads:

“Interestingly, the jump in $\tau(\theta)$ depending on the tilt angle exhibits a pronounced local minimum at $\theta \approx 25^\circ$, for all fields above H_m in this angular range. This is best seen when we plot the torque magnitude against the tilt angle, at low temperature, see Fig. 1d and Supplementary Fig. S1.”

<<

6) The last sentence in the “Magnetic torque ... transition” section needs to be made more clear. The authors should just plainly state that the component of the magnetization perpendicular to the field can be detected in magnetic torque measurements. This will avoid having to refer to prior magnetization measurements (which were perfectly reasonable) in a negative light.

>>

With the intention to highlight the difference of magnetic torque and magnetization measurements, we modified the mentioned sentence to:

“This is an additional feature revealed by our magnetic torque measurements. In comparison to the previous magnetization studies [12, 17] magnetic torque is sensitive to the transverse component of the magnetization.”

<<

7) The authors sometimes use “lfsc” and “hfsc” and other times use “lfSC” and “hfSC.” The authors need to be consistent with their usage of lower and upper case letters; not alternate between the two.

>>

In order to avoid any further confusion we now use only “lfSC” and “hfSC” throughout the text.

<<

8) Page 9 line 5. The authors should state that they are assuming that the same λ that is renormalizing the mass also determines the strength of the pairing. One has to be careful here, because there are multiple contributions (e.g. electron-electron and electron-phonon) to the mass enhancement.

>>

We agree with this comment. We added the following note on page 9:

“... (renormalized by all interactions but the pairing interaction)”

<<

9) In UPt₃ and UPd₂Al₃, it was found by way of de Haas-van Alphen measurements that the masses are just as heavy above H_m as they are below H_m . This is in contrast to what one typically finds in Ce compounds. When we consider the effective masses in UPt₃ and UPd₂Al₃, UTe₂ is no longer so unique with regards to the minimal changes in the Fermi surface at H_m .

>>

This is a very good point. In order to consider these points we added the following:

“In other systems, where quantum-oscillation measurements could be performed, such as the well-documented case CeRu₂Si₂ [41, 42] as well as the uranium systems UPt₃ [43] and UPd₂Al₃ [44], Fermi surface changes were observed across the metamagnetic field H_m , as well as heavy masses just above H_m . However, these heavy masses should be suppressed much faster by external magnetic field in cerium-based systems, which show a clearer trend to localization of the f -electrons under field and smaller Kondo temperatures than uranium systems. We will discuss later particular aspects of UTe₂ that explain why it preserves large effective masses above H_m , at least for the singular field orientations where superconductivity reappears. “

We added the following references:

[42] Flouquet, J., Haen, P., Raymond, S., Aoki, D. & Knebel, G. Itinerant metamagnetism of CeRu₂Si₂: bringing out the dead. Comparison with the new Sr₃Ru₂O₇ case. *Physica B: Condensed Matter* **319**, 251 – 261 (2002).

[43] Julian, S. R., Teunissen, P. A. A. & Wieggers, S. A. J. Fermi surface of UPt₃ from 3 to 30 T: Field-induced quasiparticle band polarization and the metamagnetic transition. *Phys. Rev. B* **46**, 9821–9824 (1992)

[44] Terashima, T. et al. Heavy fermions survive the metamagnetic transition in UPd₂Al₃. *Phys. Rev. B* **55**, 13369–13372 (1997).

<<

10) Use of the word “triggering” at the top of page 10 is strange. I would recommend replacing this with “the subject of.”

>>

We followed the suggestion of the reviewer and replaced “triggering” by “the subject of”.

<<

11) At the top of page 12, the Yamaji effect normally suppresses the interlayer conductivity in a layered system. Perhaps the authors meant “suppressed conduction” instead of “enhanced conduction?” Or maybe they are referring to the conductivity between Yamaji angles?

>>

We corrected this mistake.

<<

12) In regards to the bottom of page 13, there are papers out there still debating whether UTe₃ is spin triplet or not. Hence, they should not state that there is a consensus. Maybe they should state instead that there is consensus for UTe₃ being recognized as candidate spin triplet superconductor?

>>

The sentence was ambiguous indeed, as it mentioned phases “below H_m” when we thought of phases at low field (below 15 T), where, to the best of our knowledge, all work we know of use the spin-triplet hypothesis. In order to precise the claim, we changed this sentence into:

“In zero field, there is a consensus for UTe₂ being recognized as a candidate spin triplet superconductor with a B_{3u} or A_u symmetry.”

<<

REVIEWERS' COMMENTS

Reviewer #4 (Remarks to the Author):

I have carefully read the newly revised manuscript and the response to my previous report and I am satisfied that my previous concerns have been suitably addressed. The paper should proceed to publication.